# The Power and Limitation of Pretraining-Finetuning for Linear Regression under Covariate Shift

**Jingfeng Wu***
Johns Hopkins University
uuujf@jhu.edu

**Difan Zou***
The University of Hong Kong
dzou@cs.hku.hk

**Vladimir Braverman**
Johns Hopkins University
vova@cs.jhu.edu

**Quanquan Gu**
University of California, Los Angeles
qgu@cs.ucla.edu

**Sham M. Kakade**
Harvard University
sham@seas.harvard.edu

## Abstract

We study linear regression under covariate shift, where the marginal distribution over the input covariates differs in the source and the target domains, while the conditional distribution of the output given the input covariates is similar across the two domains. We investigate a transfer learning approach with pretraining on the source data and finetuning based on the target data (both conducted by online SGD) for this problem. We establish sharp instance-dependent excess risk upper and lower bounds for this approach. Our bounds suggest that for a large class of linear regression instances, transfer learning with $O(N^2)$ source data (and scarce or no target data) is as effective as supervised learning with $N$ target data. In addition, we show that finetuning, even with only a small amount of target data, could drastically reduce the amount of source data required by pretraining. Our theory sheds light on the effectiveness and limitation of pretraining as well as the benefits of finetuning for tackling covariate shift problems.

## 1 Introduction

In *transfer learning* [Pan and Yang, 2009, Sugiyama and Kawanabe, 2012], an algorithm is provided with abundant data from a source domain and scarce or no data from a target domain, and aims to train a model that generalizes well on the target domain. A simple yet effective approach is to *pretrain* a model with the rich source data and then *finetune* the model with the available target data via, e.g., *stochastic gradient descent* (SGD) (see, e.g., Yosinski et al. [2014]). Despite its wide applicability in practice, the power and limitation of the pretraining-finetuning based transfer learning framework is not fully understood in theory. The focus of this work is to consider this issue in a specific transfer learning setup known as *covariate shift* [Pan and Yang, 2009, Sugiyama and Kawanabe, 2012], where the source and target distributions differ in their marginal distributions over the input, but coincide in their conditional distribution of the output given the input.

Regarding the theory of learning with covariate shift, there exists a rich set of results [Ben-David et al., 2010, Germain et al., 2013, Mansour et al., 2009, Mohri and Muñoz Medina, 2012, Cortes and Mohri, 2014, Cortes et al., 2019, Kpotufe and Martinet, 2018, Hanneke and Kpotufe, 2019, Ma et al., 2022] for the (regularized) empirical risk minimizer, which minimizes the empirical loss over the source data and target data (if available) with potential regularization terms (e.g., $\ell_2$-regularization). However, in most of these works [Ben-David et al., 2010, Germain et al., 2013, Mansour et al., 2009, Mohri and Muñoz Medina, 2012, Cortes and Mohri, 2014, Cortes et al., 2019], the generalization

---

*Equal Contribution

36th Conference on Neural Information Processing Systems (NeurIPS 2022).

error on the target domain is bounded by the sum of a vanishing term (e.g., the training error) and a divergence between the two domains (see, e.g., discussions in Kpotufe and Martinet [2018], with a few notable exceptions that we will discuss later). Such bounds are very pessimistic because the additive error contributed by the source-target divergence only captures the *worst case* performance gap caused by distribution mismatch [David et al., 2010] and is too crude to describe the intriguing properties of pretraining-finetuning across different domains.

In this paper, we take a different approach to directly study the generalization performance of the pretraining-finetuning method. In particular, we consider linear regression under covariate shift, and an online SGD estimator which is firstly trained with the source data and then finetuned with the target data. We derive a target domain risk bound that is stated as a function of **(i)** the spectrum of the source and target population data covariance matrices, **(ii)** the amount of source and target data, and **(iii)** the (initial) stepsizes for pretraining and finetuning (see Theorem 3.1 for more details). Moreover, a nearly matching lower bound is provided to justify the tightness of our upper bound. The derived bounds comprehensively characterize the effects of pretraining and finetuning for *each* covariate shift problem and *each* algorithm configuration, based on which we make the following important observations:

- We compare the generalization performance (i.e., target domain excess risk) of pretraining (with source data) vs. supervised learning (with target data). We show that, for a large class of problems, $O(N^2)$ source data is sufficient for pretraining to match the performance of supervised learning with $N$ target data.

- We next show the benefits of finetuning with scarce target data. In particular, for the problem class considered before, finetuning can reduce by at least constant factors the amount of source data required by pretraining. Moreover, there exist problem instances for which the pretraining-finetuning approach requires *polynomially* less amount of total data than pretraining (with source data) or supervised learning (with target data).

- Finally, our bounds can also be applied to the supervised learning setting, i.e., linear regression with last iterate SGD. In this case, our upper bound sharpens that of Wu et al. [2021] by a logarithmic factor, and as a consequence we close the gap between the upper and lower bounds for last iterate SGD when the signal-to-noise ratio is bounded.

**Notation.** For two positive-value functions $f(x)$ and $g(x)$ we write $f(x) \lesssim g(x)$ or $f(x) \gtrsim g(x)$ if $f(x) \leq cg(x)$ or $f(x) \geq cg(x)$ for some absolute constant $c > 0$ respectively, and we write $f(x) \asymp g(x)$ if $f(x) \lesssim g(x) \lesssim f(x)$. For two vectors $\mathbf{u}$ and $\mathbf{v}$ in a Hilbert space, their inner product is denoted by $\langle \mathbf{u}, \mathbf{v} \rangle$ or equivalently, $\mathbf{u}^\top \mathbf{v}$. For a matrix $\mathbf{A}$, its spectral norm is denoted by $\|\mathbf{A}\|_2$. For two matrices $\mathbf{A}$ and $\mathbf{B}$ of appropriate dimension, their inner product is defined as $\langle \mathbf{A}, \mathbf{B} \rangle := \text{tr}(\mathbf{A}^\top \mathbf{B})$. For a positive semi-definite (PSD) matrix $\mathbf{A}$ and a vector $\mathbf{v}$ of appropriate dimension, we write $\|\mathbf{v}\|_{\mathbf{A}}^2 := \mathbf{v}^\top \mathbf{A} \mathbf{v}$. For a symmetric matrix $\mathbf{A}$ and a PSD matrix $\mathbf{B}$, we write $\|\mathbf{A}\|_{\mathbf{B}}^2 := \|\mathbf{B}^{-\frac{1}{2}} \mathbf{A} \mathbf{B}^{-\frac{1}{2}}\|_2^2$. The Kronecker/tensor product is denoted by $\otimes$. For a set $\mathbb{S}$, we use $|\mathbb{S}|$ to denote its cardinality.

## 1.1 Additional Related Work

We review some additional works that are mostly related to ours.

**Learning under Covariate Shift.** Kpotufe and Martinet [2018], Pathak et al. [2022] proposed new similarity measures to the source and target domains, and proved covariate shift bounds that do not contain an additive error of the divergence between the source and target distribution. Compared to our results, theirs can be applied to nonlinear regression/classifications as well; however in the case of linear regression, our bounds are more fine-grained and are tight upto constant factors for a broad class of problems (see Theorem 3.2), beyond being only optimal in the worst case.

It is worth noting that Hanneke and Kpotufe [2019] studied the value of target data in addressing covariate shift problems. Their discussion is based on the minimax risk bounds afforded by a given number of source and target data. In contrast, our discussion on the benefits of finetuning with target data is based on a completely different perspective, which is by comparing the *sample inflation* [Bahadur, 1967, 1971, Zou et al., 2021a] between pretraining-finetuning vs. pretraining vs. supervised learning, i.e., for *each* covariate shift problem instance, how much source (and target) data

are necessary for pretraning (and finetuning) to match the performance of supervised learning with certain amount of target data.

More recently, Ma et al. [2022] studied covariate shift problem in the nonparameteric kernel regression setting, with the assumption that the density ratio (or second moment ratio) between the target and source distribution is bounded. Their results are similar to ours in that their bounds reflect the effect of the spectrum of the source population data covariance. Since our results are dimension-free, our bounds can also be applied in the nonparametric kernel regression setting. There are two notable differences: firstly, their estimator is (weighted) ridge regression and ours is given by SGD; moreover, our results do not rely on the bounded density ratio or bounded second moment condition.

In addition, there is a vast literature on constructing more sample-efficient transfer learning algorithms, e.g., importance weighting methods [Shimodaira, 2000, Cortes et al., 2010] and learning invariant representations [Arjovsky et al., 2019, Wu et al., 2019], to mention a few. Along this line, Lei et al. [2021] proposed nearly minimax optimal estimator for linear regression under distribution shift, but their method relies on the knowledge of target population covariance matrix. Developing new transfer learning algorithms is beyond the agenda in this paper.

**SGD.** The pretraining and finetuning discussed in this work are both conducted by online SGD, therefore our results are closely related to the generalization analysis of online SGD for linear regression in the supervised learning context [Bach and Moulines, 2013, Dieuleveut et al., 2017, Jain et al., 2017a,b, Ge et al., 2019, Zou et al., 2021b, Varre et al., 2021, Wu et al., 2021]. From a technical point of view, our theoretical results can be viewed as an extension of the SGD analysis from the supervised learning setting to the covariate shift setting.

## 2  Problem Setup

**Transfer Learning.** We use $\mathbf{x}$ to denote a covariate in a Hilbert space (that can be $d$-dimensional or countably infinite dimensional), and $y \in \mathbb{R}$ to denote its response. Consider a source and a target data distribution, denoted by $\mathcal{D}_{\texttt{source}}$ and $\mathcal{D}_{\texttt{target}}$ respectively. In the problem of *transfer learning*, we are given with $M$ data sampled independently from the source distribution, and $N$ data sampled independently from the target distribution (where $N \ll M$ or even $N = 0$), denoted by

$$(\mathbf{x}_i, y_i)_{i=1}^{M+N}, \text{ where } (\mathbf{x}_i, y_i) \sim \begin{cases} \mathcal{D}_{\texttt{source}}, & i = 1, \dots, M; \\ \mathcal{D}_{\texttt{target}}, & i = M+1, \dots, M+N. \end{cases}$$

The goal of transfer learning is to learn a model based on the $M + N$ data that can generalize on the target domain. We are particularly interested in the *covariate shift* problem in transfer learning, where the source and target distributions satisfy: $\mathcal{D}_{\texttt{source}}(y|\mathbf{x}) = \mathcal{D}_{\texttt{target}}(y|\mathbf{x})$ but $\mathcal{D}_{\texttt{source}}(\mathbf{x}) \neq \mathcal{D}_{\texttt{target}}(\mathbf{x})$.

**Linear Regression under Covariate Shift.** A covariate shift problem is formally defined in the context of linear regression by Definitions 1 and 2.

**Definition 1** (Covariances conditions)**.** Assume that each entry and the trace of the source and target data covariance matrices are finite. Denote the source and target data covariance matrices by

$$\mathbf{G} := \mathbb{E}_{\mathcal{D}_{\texttt{source}}}[\mathbf{x}\mathbf{x}^\top], \qquad \mathbf{H} := \mathbb{E}_{\mathcal{D}_{\texttt{target}}}[\mathbf{x}\mathbf{x}^\top],$$

respectively, and denote their eigenvalues by $(\mu_i)_{i \geq 1}$ and $(\lambda_i)_{i \geq 1}$, respectively. For convenience assume that both $\mathbf{G}$ and $\mathbf{H}$ are strictly positive definite.

**Definition 2** (Model conditions)**.** For a parameter $\mathbf{w}$, define its source and target risks by

$$\texttt{Risk}_{\texttt{source}}(\mathbf{w}) := \frac{1}{2}\mathbb{E}_{\mathcal{D}_{\texttt{source}}}(y - \mathbf{w}^\top\mathbf{x})^2, \quad \texttt{Risk}_{\texttt{target}}(\mathbf{w}) := \frac{1}{2}\mathbb{E}_{\mathcal{D}_{\texttt{target}}}(y - \mathbf{w}^\top\mathbf{x})^2,$$

respectively. Assume that there is a parameter $\mathbf{w}^*$ that *simultaneously* minimizes both source and target risks, i.e., $\mathbf{w}^* \in \arg\min_{\mathbf{w}} \texttt{Risk}_{\texttt{source}}(\mathbf{w}) \cap \arg\min_{\mathbf{w}} \texttt{Risk}_{\texttt{target}}(\mathbf{w})$. For convenience assume that $\mathbf{w}^*$ is unique.

We remark that the strict positive definiteness of $\mathbf{G}$ and $\mathbf{H}$ in Definition 1 and the uniqueness of $\mathbf{w}^*$ in Definition 2 are only made for the ease of presentation. Otherwise one can set $\mathbf{w}^*$ to be the minimum-norm solution, i.e., $\mathbf{w}^* = \arg\min\{\|\mathbf{w}\|_2 : \mathbf{w} \in \arg\min_{\mathbf{w}} \texttt{Risk}_{\texttt{source}}(\mathbf{w}) \cap$

$\arg\min_{\mathbf{w}} \texttt{Risk}_{\texttt{target}}(\mathbf{w})\}$, and our results still hold. This argument also holds in a reproducing kernel Hilbert space [Schölkopf et al., 2002].

Our definitions of linear regression under covaraite shift follow from Lei et al. [2021], Ma et al. [2022]. In the literature, covariate shift problem often assumes the source and target distributions differ in their marginal distributions over the input, but coincide in their conditional distribution of the output given the input (see, e.g., Sections 1.3.3 and 1.3.4 in Sugiyama et al. [2012]). When applied to the well-specified linear regression models, i.e., $\mathbb{E}[y|\mathbf{x}] = \mathbf{x}^\top \mathbf{w}^*$ for some parameter $\mathbf{w}^*$, the latter condition turns out to require the optimal parameter $\mathbf{w}^*$ is identical for both source and target domains. Therefore, the problems described by Definitions 1 and 2 include at least all well-specified linear regression problems under standard covariate shift conditions.

**Excess Risk.** For linear regression under covariate shift, the performance of a parameter $\mathbf{w}$ is measured by its *target domain excess risk*, i.e.,

$$\texttt{ExcessRisk}(\mathbf{w}) := \texttt{Risk}_{\texttt{target}}(\mathbf{w}) - \texttt{Risk}_{\texttt{target}}(\mathbf{w}^*) = \frac{1}{2}\langle \mathbf{H}, \ (\mathbf{w} - \mathbf{w}^*) \otimes (\mathbf{w} - \mathbf{w}^*)\rangle.$$

**SGD.** The transfer learning algorithm of our interests is pretraining-finetuning via *online stochastic gradient descent with geometrically decaying stepsizes*[2] (SGD). Without lose of generality, we assume the SGD iterates are initialized from $\mathbf{w}_0 = \mathbf{0}$. Then the SGD iterates are sequentially updated as follows:

$$\mathbf{w}_t = \mathbf{w}_{t-1} - \gamma_{t-1}(\mathbf{x}_t \mathbf{x}_t^\top \mathbf{w}_{t-1} - \mathbf{x}_t \mathbf{y}_t), \quad t = 1, \dots, M + N,$$
$$\text{where } \gamma_t = \begin{cases} \gamma_0/2^\ell, & 0 \le t < M, \ \ell = \lfloor t/\log(M)\rfloor; \\ \gamma_M/2^\ell, & M \le t < N, \ \ell = \lfloor (t-M)/\log(N)\rfloor, \end{cases} \tag{SGD}$$

and the output is the last iterate, i.e., $\mathbf{w}_{M+N}$. Here $\gamma_0$ and $\gamma_M$ are two hyperparameters that correspond to the initial stepsizes for pretraining and finetuning, respectively. In both pretraining and finetuning phases, the stepsize scheduler in (SGD) is epoch-wisely a constant and decays geometrically every certain number of epochs, which is widely used in deep learning [He et al., 2015]. We note that such (SGD) for linear regression has been analyzed by Ge et al. [2019], Wu et al. [2021] in the context of supervised learning. Our goal in this work is to understand the generalization of (SGD) in the covariate shift problems.

**Assumptions.** The following assumptions [Zou et al., 2021b, Wu et al., 2021] are crucial in our analysis.

**Assumption 1** (Fourth moment conditions). *Assume that for both source and target distribution the fourth moment of the covariates is finite. Moreover:*

A *There is a constant $\alpha > 0$ such that for every PSD matrix $\mathbf{A}$ it holds that*

$$\mathbb{E}_{\mathcal{D}_{\texttt{source}}}[\mathbf{x}\mathbf{x}^\top \mathbf{A}\mathbf{x}\mathbf{x}^\top] \preceq \alpha \cdot \text{tr}(\mathbf{G}\mathbf{A}) \cdot \mathbf{G}, \quad \mathbb{E}_{\mathcal{D}_{\texttt{target}}}[\mathbf{x}\mathbf{x}^\top \mathbf{A}\mathbf{x}\mathbf{x}^\top] \preceq \alpha \cdot \text{tr}(\mathbf{H}\mathbf{A}) \cdot \mathbf{H}.$$

*Clearly, it must hold that $\alpha \ge 1$.*

B *There is a constant $\beta > 0$ such that for every PSD matrix $\mathbf{A}$ it holds that*

$$\mathbb{E}_{\mathcal{D}_{\texttt{source}}}[\mathbf{x}\mathbf{x}^\top \mathbf{A}\mathbf{x}\mathbf{x}^\top] - \mathbf{G}\mathbf{A}\mathbf{G} \succeq \beta \cdot \text{tr}(\mathbf{G}\mathbf{A}) \cdot \mathbf{G}, \quad \mathbb{E}_{\mathcal{D}_{\texttt{target}}}[\mathbf{x}\mathbf{x}^\top \mathbf{A}\mathbf{x}\mathbf{x}^\top] - \mathbf{H}\mathbf{A}\mathbf{H} \succeq \beta \cdot \text{tr}(\mathbf{H}\mathbf{A}) \cdot \mathbf{H}.$$

Assumption 1 holds with $\alpha = 3$ and $\beta = 1$ given that $\mathcal{D}_{\texttt{source}}(\mathbf{x}) = \mathcal{N}(\mathbf{0}, \mathbf{G})$ and $\mathcal{D}_{\texttt{target}}(\mathbf{x}) = \mathcal{N}(\mathbf{0}, \mathbf{H})$. Moreover, Assumption 1A holds if both $\mathbf{H}^{-\frac{1}{2}} \cdot \mathcal{D}_{\texttt{source}}(\mathbf{x})$ and $\mathbf{G}^{-\frac{1}{2}} \cdot \mathcal{D}_{\texttt{target}}(\mathbf{x})$ have sub-Gaussian tails [Zou et al., 2021b]. For more exemplar distributions that satisfy Assumption 1, we refer the reader to Wu et al. [2021].

**Assumption 2** (Noise condition). *Assume that there is a constant $\sigma^2 > 0$ such that*

$$\mathbb{E}_{\mathcal{D}_{\texttt{source}}}[(y - \langle \mathbf{w}^*, \mathbf{x}\rangle)^2 \mathbf{x}\mathbf{x}^\top] \preceq \sigma^2 \cdot \mathbf{G}, \quad \mathbb{E}_{\mathcal{D}_{\texttt{target}}}[(y - \langle \mathbf{w}^*, \mathbf{x}\rangle)^2 \mathbf{x}\mathbf{x}^\top] \preceq \sigma^2 \cdot \mathbf{H}.$$

Assumption 2 puts mild requirements on the conditional distribution of the response given input covariates for both source and target distribution. In particular, Assumption 2 is directly implied by the following Assumption 2' for a well-specified linear regression model under covariate shift.

---

[2]For the conciseness of presentation we focus on SGD with geometrically decaying stepsizes. With the provided techniques, our results can be easily extended to SGD with tail geometrically decaying stepsizes [Wu et al., 2021] as well.

**Assumption 2'** (Well-specified noise). *Assume that for both source and target distributions, the response (conditional on input covariates) is given by*

$$y = \mathbf{x}^\top \mathbf{w}^* + \epsilon, \text{ where } \epsilon \sim \mathcal{N}(0, \sigma^2) \text{ and } \epsilon \text{ is independent with } \mathbf{x}.$$

**Additional Notation.** Let $\mathbb{N}_+ := \{1, 2, \dots\}$. For an index set $\mathbb{K} \subset \mathbb{N}_+$, its complement is defined by $\mathbb{K}^c := \mathbb{N}_+ - \mathbb{K}$. Then for an index set $\mathbb{K} \subset \mathbb{N}_+$ and a scalar $a \geq 0$, we define

$$\mathbf{H}_\mathbb{K} := \sum_{i \in \mathbb{K}} \lambda_i \mathbf{v}_i \mathbf{v}_i^\top, \quad \mathbf{H}_\mathbb{K}^{-1} := \sum_{i \in \mathbb{K}} \frac{1}{\lambda_i} \mathbf{v}_i \mathbf{v}_i^\top, \quad a\mathbf{I}_\mathbb{K} + \mathbf{H}_{\mathbb{K}^c} := \sum_{i \in \mathbb{K}} a\mathbf{v}_i \mathbf{v}_i + \sum_{i \notin \mathbb{K}} \lambda_i \mathbf{v}_i \mathbf{v}_i,$$

where $(\lambda_i)_{i \geq 1}$ and $(\mathbf{v}_i)_{i \geq 1}$ are corresponding eigenvalues and eigenvectors of $\mathbf{H}$. One can verify that $\mathbf{H}_\mathbb{K}^{-1}$ is equivalent to the (pseudo) inverse of $\mathbf{H}_\mathbb{K}$. Similarly, we define $\mathbf{G}_\mathbb{J}$, $\mathbf{G}_\mathbb{J}^{-1}$ and $a\mathbf{I}_\mathbb{J} + \mathbf{G}_{\mathbb{J}^c}$ according to the eigenvalues and eigenvectors of $\mathbf{G}$.

## 3 Main Results

**An Upper Bound.** We begin with presenting an upper bound for the target domain excess risk achieved by the pretraining-finetuning method.

**Theorem 3.1** (upper bound). *Suppose that Assumptions 1A and 2 hold. Let $\mathbf{w}_{M+N}$ be the output of (SGD). Let $M_{\text{eff}} := M/\log(M)$, $N_{\text{eff}} := N/\log(N)$. Suppose that $\gamma_0, \gamma_M < \min\{1/(4\alpha \operatorname{tr}(\mathbf{G})), 1/(4\alpha \operatorname{tr}(\mathbf{H}))\}$. Then it holds that*

$$\texttt{ExcessRisk}(\mathbf{w}_{M+N}) \leq \texttt{BiasError} + \texttt{VarError}.$$

*Moreover, for any two index sets $\mathbb{J}, \mathbb{K} \subset \mathbb{N}_+$, it holds that*

$$\texttt{VarError} \lesssim \sigma^2 \cdot \left( \frac{D_{\text{eff}}^{\texttt{finetune}}}{M_{\text{eff}}} + \frac{D_{\text{eff}}}{N_{\text{eff}}} \right);$$

$$\texttt{BiasError} \lesssim \left\| \prod_{t=M}^{M+N-1}(\mathbf{I} - \gamma_t \mathbf{H}) \prod_{t=0}^{M-1}(\mathbf{I} - \gamma_t \mathbf{G})(\mathbf{w}_0 - \mathbf{w}^*) \right\|_\mathbf{H}^2$$

$$+ \alpha \cdot \left\| \mathbf{w}_0 - \mathbf{w}^* \right\|_{\frac{\mathbf{I}_\mathbb{J}}{M_{\text{eff}}\gamma_0} + \mathbf{G}_{\mathbb{J}^c}}^2 \cdot \frac{D_{\text{eff}}^{\texttt{finetune}}}{M_{\text{eff}}}$$

$$+ \alpha \cdot \left( \left\| \prod_{t=0}^{M-1}(\mathbf{I} - \gamma_t \mathbf{G})(\mathbf{w}_0 - \mathbf{w}^*) \right\|_{\frac{\mathbf{I}_\mathbb{K}}{N_{\text{eff}}\gamma_M} + \mathbf{H}_{\mathbb{K}^c}}^2 + \left\| \mathbf{w}_0 - \mathbf{w}^* \right\|_{\frac{\mathbf{I}_\mathbb{J}}{M_{\text{eff}}\gamma_0} + \mathbf{G}_{\mathbb{J}^c}}^2 \right) \cdot \frac{D_{\text{eff}}}{N_{\text{eff}}},$$

*where*

$$D_{\text{eff}}^{\texttt{finetune}} := \operatorname{tr}\left( \prod_{t=0}^{N-1}(\mathbf{I} - \gamma_{M+t}\mathbf{H})^2 \mathbf{H} \cdot (\mathbf{G}_\mathbb{J}^{-1} + M_{\text{eff}}^2 \gamma_0^2 \cdot \mathbf{G}_{\mathbb{J}^c}) \right),$$
$$D_{\text{eff}} := |\mathbb{K}| + N_{\text{eff}}^2 \gamma_M^2 \cdot \sum_{i \notin \mathbb{K}} \lambda_i^2. \tag{1}$$

*In particular, the upper bounds are optimized when*

$$\mathbb{J} = \{j : \mu_j \geq 1/(\gamma_0 M_{\text{eff}})\}, \quad \mathbb{K} = \{k : \lambda_k \geq 1/(\gamma_M N_{\text{eff}})\}. \tag{2}$$

The upper bound in Theorem 3.1 contains a bias error stemming from the incorrect initialization $\mathbf{w}_0 \neq \mathbf{w}^*$, and a variance error caused by the additive label noise $y - \mathbf{x}^\top \mathbf{w}^* \neq 0$. In particular, $M_{\text{eff}}$ and $N_{\text{eff}}$ are the *effective number of source and target data*, respectively, due to the effect of the geometrically decaying stepsizes in (SGD). Moreover, $D_{\text{eff}}$ can be regarded as the *effective dimension of supervised learning* [Wu et al., 2021] and $D_{\text{eff}}^{\texttt{finetune}}$ can be regared as the *effective dimension of pretraining-finetuning*. Note that $D_{\text{eff}}^{\texttt{finetune}}$ is determined jointly by the spectrum of the source and target population covariance matrices as well as the stepsizes for pretraining and finetuning.

To better illustrate the spirit of Theorem 3.1, let us consider an example where $\|\mathbf{w}_0 - \mathbf{w}^*\|_2^2, \sigma^2 \lesssim 1$, $\gamma_0 \asymp 1$, and $\operatorname{tr}(\mathbf{G}) \asymp \operatorname{tr}(\mathbf{H}) \asymp 1$ (so that the spectrum of $\mathbf{G}$ and $\mathbf{H}$ must decay fast), then the bound in Theorem 3.1 vanishes provided that

$$D_{\text{eff}} = o(N_{\text{eff}}), \quad D_{\text{eff}}^{\texttt{finetune}} = o(M_{\text{eff}}). \tag{3}$$

For the first condition in (3) to happen one needs

$$|\mathbb{K}| = o(N/\log N), \quad \gamma_M^2 \cdot \sum_{i \notin \mathbb{K}} \lambda_i^2 = o(\log N/N),$$

which can be satisfied when **(i)** the number of target data $N$ is large and the finetuning stepsize $\gamma_M \asymp 1$, or when **(ii)** $N$ is small and $\gamma_M$ is also small (which can depend on $N$). The second condition in (3) can happen under various situations, e.g., when **(i)** $N$ is large and $\gamma_M \asymp 1$, or when **(ii)** $N$, $\gamma_M$ are small but the amount of source data $M$ is large and that

$$\mathrm{tr}(\mathbf{H}\mathbf{G}_{\mathbb{J}}^{-1}) = o(M/\log M), \quad \mathrm{tr}(\mathbf{H}\mathbf{G}_{\mathbb{J}^c}) = o(\log M/M),$$

which will hold when $\mathbf{G}$ aligns well with $\mathbf{H}$ (as a sanity check these hold automatically when $\mathbf{G} = \mathbf{H}$ and $M$ is large). To summarize, in case **(i)** the amount of target data is plentiful so that finetuning with large stepsize leads to generalization (which is essentially supervised learning); and in case **(ii)**, even though the target data is scarce, pretraining with abundant source data can still generalize given that the source and target population covariance matrices are well aligned.

**A Lower Bound.** The following theorem provides a nearly matching lower bound.

**Theorem 3.2** (lower bound). *Suppose that Assumptions 1B and 2' hold. Let $\mathbf{w}_{M+N}$ be the output of* (SGD). *Let $M_{\mathtt{eff}} := M/\log(M)$, $N_{\mathtt{eff}} := N/\log(N)$, and suppose that $M_{\mathtt{eff}}, N_{\mathtt{eff}} \geq 10$. Suppose that $\gamma_0 < 1/\|\mathbf{G}\|_2$, $\gamma_M < 1/\|\mathbf{H}\|_2$. Then it holds that*

$$\mathtt{ExcessRisk}(\mathbf{w}_{M+N}) = \mathtt{BiasError} + \mathtt{VarError}.$$

*Moreover, for the index sets $\mathbb{K}$ and $\mathbb{J}$ defined in (2), it holds that*

$$\mathtt{VarError} \gtrsim \sigma^2 \cdot \left( \frac{D_{\mathtt{eff}}^{\mathtt{finetune}}}{M_{\mathtt{eff}}} + \frac{D_{\mathtt{eff}}}{N_{\mathtt{eff}}} \right);$$

$$\mathtt{BiasError} \gtrsim \left\| \textstyle\prod_{t=M}^{M+N-1}(\mathbf{I} - \gamma_t\mathbf{H}) \prod_{t=0}^{M-1}(\mathbf{I} - \gamma_t\mathbf{G})(\mathbf{w}_0 - \mathbf{w}^*) \right\|_{\mathbf{H}}^2$$

$$+ \beta \cdot \|\mathbf{w}_0 - \mathbf{w}^*\|_{\mathbf{G}_{\mathbb{J}^c}}^2 \cdot \frac{D_{\mathtt{eff}}^{\mathtt{finetune}}}{M_{\mathtt{eff}}} + \beta \cdot \left\| \textstyle\prod_{t=0}^{M-1}(\mathbf{I} - \gamma_t\mathbf{G})(\mathbf{w}_0 - \mathbf{w}^*) \right\|_{\mathbf{H}_{\mathbb{K}^c}}^2 \cdot \frac{D_{\mathtt{eff}}}{N_{\mathtt{eff}}},$$

*where $D_{\mathtt{eff}}$ and $D_{\mathtt{eff}}^{\mathtt{finetune}}$ are as defined in (1).*

The lower bound in Theorem 3.2 suggests that the upper bound in Theorem 3.1 is tight upto constant factor in terms of variance error, and is also tight in terms of bias error except for the following additional parts in the respective places:

$$\left\| \textstyle\prod_{t=0}^{M-1}(\mathbf{I} - \gamma_t\mathbf{G})(\mathbf{w}_0 - \mathbf{w}^*) \right\|_{\frac{\mathbf{I}_{\mathbb{K}}}{N_{\mathtt{eff}}\gamma_M}}^2, \quad \|\mathbf{w}_0 - \mathbf{w}^*\|_{\frac{\mathbf{I}_{\mathbb{J}}}{M_{\mathtt{eff}}\gamma_0}}^2, \quad \|\mathbf{w}_0 - \mathbf{w}^*\|_{\frac{\mathbf{I}_{\mathbb{J}}}{M_{\mathtt{eff}}\gamma_0} + \mathbf{G}_{\mathbb{J}^c}}^2.$$

In particular, the upper and lower bounds match ignoring constant factors provided that

$$\|\mathbf{w}_0 - \mathbf{w}^*\|_{\mathbf{G}}^2 \lesssim \sigma^2, \quad \left\| \textstyle\prod_{t=0}^{M-1}(\mathbf{I} - \gamma_t\mathbf{G})(\mathbf{w}_0 - \mathbf{w}^*) \right\|_{\mathbf{H}}^2 \lesssim \sigma^2,$$

which hold in a statistically interesting regime where the signal-to-noise ratios, $\|\mathbf{w}_0 - \mathbf{w}^*\|_{\mathbf{G}}^2/\sigma^2$, $\|\mathbf{w}_0 - \mathbf{w}^*\|_{\mathbf{H}}^2/\sigma^2$, are bounded and $\mathbf{G}$ commutes with $\mathbf{H}$.

**Implication for Pretraining.** If target data is unavailable, Theorems 3.1 and 3.2 imply the following corollary for pretraining.

**Corollary 3.3** (Learning with only source data). *Suppose that Assumptions 1A and 2 hold. Let $\mathbf{w}_{M+0}$ be the output of* (SGD) *with $M > 100$ source data and 0 target data. Suppose that $\gamma := \gamma_0 < 1/(4\alpha\,\mathrm{tr}(\mathbf{H}))$. Then it holds that*

$$\mathtt{ExcessRisk}(\mathbf{w}_{M+0}) \lesssim \left\| \textstyle\prod_{t=0}^{M-1}(\mathbf{I} - \gamma_t\mathbf{G})(\mathbf{w}_0 - \mathbf{w}^*) \right\|_{\mathbf{H}}^2 + \left( \alpha\|\mathbf{w}_0 - \mathbf{w}^*\|_{\frac{\mathbf{I}_{\mathbb{J}}}{M_{\mathtt{eff}}\gamma} + \mathbf{G}_{\mathbb{J}^c}}^2 + \sigma^2 \right) \frac{D_{\mathtt{eff}}^{\mathtt{pretrain}}}{M_{\mathtt{eff}}},$$

*where $D_{\mathtt{eff}}^{\mathtt{pretrain}} := \mathrm{tr}(\mathbf{H}\mathbf{G}_{\mathbb{J}}^{-1}) + M_{\mathtt{eff}}^2\gamma^2 \cdot \mathrm{tr}(\mathbf{H}\mathbf{G}_{\mathbb{J}^c})$ and $\mathbb{J} \subset \mathbb{N}_+$ can be any index set. If in addition Assumptions 1B and 2' hold, then for the index set $\mathbb{J}$ defined in (2), it holds that*

$$\mathtt{ExcessRisk}(\mathbf{w}_{M+0}) \gtrsim \left\| \textstyle\prod_{t=0}^{M-1}(\mathbf{I} - \gamma_t\mathbf{G})(\mathbf{w}_0 - \mathbf{w}^*) \right\|_{\mathbf{H}}^2 + \left( \beta\|\mathbf{w}_0 - \mathbf{w}^*\|_{\mathbf{G}_{\mathbb{J}^c}}^2 + \sigma^2 \right) \cdot \frac{D_{\mathtt{eff}}^{\mathtt{pretrain}}}{M_{\mathtt{eff}}}.$$

Corollary 3.3 sharply characterizes the generalization of pretraining method, and is tight upto constant factors provided with a bounded signal-to-noise ratio, i.e., $\|\mathbf{w}_0 - \mathbf{w}^*\|_{\mathbf{G}}^2 \lesssim \sigma^2$. Corollary 3.3 can be interpreted in a similar way as Theorem 3.1. Moreover, these sharp bounds for pretraining and pretraining-finetuning enable us to study the effects of pretraining and finetuning thoroughly, which we will do in Section 4.

**Implication for Supervised Learning.** As a bonus, we can also apply Theorems 3.1 and 3.2 in the setting of supervised learning.

**Corollary 3.4** (Learning with only target data). *Suppose that Assumptions 1A and 2 hold. Let $\mathbf{w}_{0+N}$ be the output of (SGD) with 0 source data and $N > 100$ target data. Suppose that $\gamma := \gamma_M < 1/(4\alpha(\mathrm{tr}(\mathbf{G})))$. Then it holds that*

$$\texttt{ExcessRisk}(\mathbf{w}_{0+N}) \lesssim \left\| \prod_{t=0}^{N-1}(\mathbf{I} - \gamma_t \mathbf{H})(\mathbf{w}_0 - \mathbf{w}^*) \right\|_{\mathbf{H}}^2 + \left( \alpha \|\mathbf{w}_0 - \mathbf{w}^*\|^2_{\frac{\mathbf{I}_{\mathbb{K}}}{N_{\texttt{eff}}\gamma}+\mathbf{H}_{\mathbb{K}^c}} + \sigma^2 \right) \frac{D_{\texttt{eff}}}{N_{\texttt{eff}}},$$

*where $D_{\texttt{eff}}$ is as defined in (1) and $\mathbb{K} \subset \mathbb{N}_+$ can be any index set. If in addition Assumptions 1B and 2' hold, then for the index set $\mathbb{K}$ defined in (2), it holds that*

$$\texttt{ExcessRisk}(\mathbf{w}_{0+N}) \gtrsim \left\| \prod_{t=0}^{N-1}(\mathbf{I} - \gamma_t \mathbf{H})(\mathbf{w}_0 - \mathbf{w}^*) \right\|_{\mathbf{H}}^2 + \left( \beta \|\mathbf{w}_0 - \mathbf{w}^*\|^2_{\mathbf{H}_{\mathbb{K}^c}} + \sigma^2 \right) \cdot \frac{D_{\texttt{eff}}}{N_{\texttt{eff}}}.$$

We remark that the upper bound in Corollary 3.4 improves the related bound in Wu et al. [2021] by a logarithmic factor, and matches the lower bound upto constant factors given that $\|\mathbf{w}_0 - \mathbf{w}^*\|_{\mathbf{H}}^2 \lesssim \sigma^2$.

For a more detailed comparison between the bounds in Theorem 3.1, Corollaries 3.3 and 3.4, we refer the reader to Table 1 in Appendix A.

# 4 Discussions

With the established bounds, we are ready to discuss the power and limitation of pretraining and finetuning by comparing them to supervised learning.

**The Power of Pretraining.** For covariate shift problem, pretraining with infinite many source data can learn the true model. But when there are only a finite number of source data, it is unclear how the effect of pretraining compares to the effect of supervised learning (with finite many target data). Our next result quantitatively address this question by comparing Corollary 3.3 with Corollary 3.4.

**Theorem 4.1** (Pretraining vs. supervised learning). *Suppose that Assumptions 1 and 2' hold. Let $\mathbf{w}_{0+N^{\texttt{sl}}}$ be the output of (SGD) with optimally tuned initial stepsize, 0 source data and $N^{\texttt{sl}} > 100$ target data. Let $\mathbf{w}_{M+0}$ be the output of (SGD) with optimally tuned initial stepsize, $M$ source data and 0 target data. Let $M_{\texttt{eff}} := M/\log(M)$, $N^{\texttt{sl}}_{\texttt{eff}} := N^{\texttt{sl}}/\log(N^{\texttt{sl}})$. Suppose all SGD methods are initialized from $\mathbf{0}$. Then for every covariate shift problem instance $(\mathbf{w}^*, \mathbf{H}, \mathbf{G})$ such that $\mathbf{H}, \mathbf{G}$ commute, it holds that*

$$\texttt{ExcessRisk}(\mathbf{w}_{M+0}) \lesssim (1 + \alpha \|\mathbf{w}^*\|_{\mathbf{G}}^2/\sigma^2) \cdot \texttt{ExcessRisk}(\mathbf{w}_{0+N^{\texttt{sl}}})$$

*provided that*

$$M_{\texttt{eff}} \geq (N^{\texttt{sl}}_{\texttt{eff}})^2 \cdot \frac{4\|\mathbf{H}_{\mathbb{K}^*}\|_{\mathbf{G}}}{\alpha D^{\texttt{sl}}_{\texttt{eff}}},$$

*where*

$$\mathbb{K}^* := \{k : \lambda_k \geq 1/(N^{\texttt{sl}}_{\texttt{eff}}\gamma^{\texttt{sl}})\}, \quad D^{\texttt{sl}}_{\texttt{eff}} := |\mathbb{K}^*| + (N^{\texttt{sl}}_{\texttt{eff}}\gamma^{\texttt{sl}})^2 \sum_{i \notin \mathbb{K}^*} \lambda_i^2,$$

*and $\gamma^{\texttt{sl}} < 1/\|\mathbf{H}\|_2$ refers to the optimal initial stepsize for supervised learning.*

We now explain the implication of Theorem 4.1. First of all, it is of statistical interest to consider a signal-to-noise ratio bounded from above, i.e., $\|\mathbf{w}^*\|_{\mathbf{G}}^2/\sigma^2 \lesssim 1$. Note that $D^{\texttt{sl}}_{\texttt{eff}} \geq 1$ when $N^{\texttt{sl}}$ is large. Moreover, recall that $|\mathbb{K}^*| \leq D^{\texttt{sl}}_{\texttt{eff}} = o(N^{\texttt{sl}}_{\texttt{eff}})$ when supervised learning can achieve a vanishing excess risk. Finally, $\|\mathbf{H}_{\mathbb{K}^*}\|_{\mathbf{G}} := \|\mathbf{G}^{-\frac{1}{2}}\mathbf{H}_{\mathbb{K}^*}\mathbf{G}^{-\frac{1}{2}}\|_2 \lesssim 1$ can be satisfied if the top $|\mathbb{K}^*| = o(N^{\texttt{sl}}_{\texttt{eff}})$ eigenvalues subspace of $\mathbf{H}$ mostly falls into the top eigenvalues subspace of $\mathbf{G}$. Under these remarks, Theorem 4.1 suggests that: in the bounded signal-to-noise cases, pretraining with $O(N^2)$ source data is *no worse than* supervised learning with $N$ target data (ignoring constant factors), for *every* covariate shift problem such that the *top* eigenvalues subspace of the target covariance matrix aligns well with that of the source covariance matrix.

**The Power of Pretraining-Finetuning.** We next discuss the effect of pretraining-finetuning by comparing Theorem 3.1 with Corollary 3.4.

**Theorem 4.2** (Pretraining-finetuning vs. supervised learning). *Suppose that Assumptions 1 and 2' hold. Let $\mathbf{w}_{0+N^{\texttt{sl}}}$ be the output of (SGD) with optimally tuned initial stepsize, 0 source data and $N^{\texttt{sl}} > 100$ target data. Let $\mathbf{w}_{M+N}$ be the output of (SGD) with optimally tuned initial stepsize,*

*$M$ source data and $N$ target data. Let $M_{\mathtt{eff}} := M/\log(M)$, $N_{\mathtt{eff}} := N/\log(N)$, $N_{\mathtt{eff}}^{\mathtt{sl}} := N^{\mathtt{sl}}/\log(N^{\mathtt{sl}})$. Suppose all SGD methods are initialized from $\mathbf{0}$. Then for every covariate shift problem instance $(\mathbf{w}^*, \mathbf{H}, \mathbf{G})$ such that $\mathbf{H}, \mathbf{G}$ commute, it holds that*

$$\mathtt{ExcessRisk}(\mathbf{w}_{M+N}) \lesssim (1 + \alpha\|\mathbf{w}^*\|_{\mathbf{G}}^2/\sigma^2) \cdot \mathtt{ExcessRisk}(\mathbf{w}_{0+N^{\mathtt{sl}}})$$

*provided that*

$$M_{\mathtt{eff}} \geq (N_{\mathtt{eff}}^{\mathtt{sl}})^2 \cdot \frac{4\|\mathbf{H}_{\mathbb{K}^\dagger}\|_{\mathbf{G}}}{\alpha D_{\mathtt{eff}}^{\mathtt{sl}}},$$

*where*

$$\mathbb{K}^\dagger := \{k : N_{\mathtt{eff}}^{\mathtt{sl}} \log(N_{\mathtt{eff}}^{\mathtt{sl}}) \operatorname{tr}(\mathbf{H})/(N_{\mathtt{eff}} D_{\mathtt{eff}}^{\mathtt{sl}}) > \lambda_k \geq 1/(N_{\mathtt{eff}}^{\mathtt{sl}} \gamma^{\mathtt{sl}})\} \subset \mathbb{K}^*,$$

*and $\mathbb{K}^*$, $D_{\mathtt{eff}}^{\mathtt{sl}}$, $\gamma^{\mathtt{sl}}$ are as defined in Theorem 4.1.*

Theorem 4.2 can be interpreted in a similar way as Theorem 4.1. The only difference is that Theorem 4.2 puts a *milder* condition regarding the alignment of $\mathbf{H}$ and $\mathbf{G}$ than Theorem 4.1. In particular, Theorem 4.2 only requires the "middle" eigenvalues subspace of $\mathbf{H}$ mostly falls into the top eigenvalues subspace of $\mathbf{G}$. Moreover, the index set $\mathbb{K}^\dagger$ shrinks (hence $\|\mathbf{H}_{\mathbb{K}^\dagger}\|_{\mathbf{G}}$ decreases) as the number of target data for finetuning increases. This indicates that finetuning can help save the amount of source data for pretraining.

**The Limitation of Pretraining vs. the Power of Finetuning.** The following example further demonstrates the limitation of pretraining and the power of finetuning.

**Example 4.3** (Pretraining-finetuning vs. pretraining vs. supervised learning). *Let $\epsilon > 0$ be a sufficiently small constant. Consider a covariate shift problem instance given by*

$$\mathbf{w}^* = (1, 1, 0, 0, \dots)^\top, \quad \sigma^2 = 1,$$
$$\mathbf{H} = \operatorname{diag}(1, \underbrace{\epsilon^{0.5}, \dots, \epsilon^{0.5}}_{2\epsilon^{-0.5} \text{ copies}}, 0, 0, \dots), \quad \mathbf{G} = \operatorname{diag}(\epsilon^2, 1, 0, 0, \dots).$$

*One may verify that $\operatorname{tr}(\mathbf{H}) \asymp \operatorname{tr}(\mathbf{G}) \asymp 1$ and that $\|\mathbf{w}^*\|_{\mathbf{H}}^2 \asymp \|\mathbf{w}^*\|_{\mathbf{G}}^2 \asymp \sigma^2 \asymp 1$. The following holds for the (SGD) output:*

- *supervised learning: for $\mathtt{ExcessRisk}(\mathbf{w}_{0+N}) < \epsilon$, it is necessary to have that $N \gtrsim \epsilon^{-1.5}$;*

- *pretraining: for $\mathtt{ExcessRisk}(\mathbf{w}_{M+0}) < \epsilon$, it is necessary to have that $M \gtrsim \epsilon^{-2}$;*

- *pretrain-finetuning: for $\mathtt{ExcessRisk}(\mathbf{w}_{M+N}) < \epsilon$, it suffices to set $\gamma_0 \asymp 1$, $\gamma_M \asymp \epsilon$, and $M \asymp \epsilon^{-1} \log(\epsilon^{-1})$, $N \asymp \epsilon^{-1} \log^2(\epsilon^{-1})$.*

It is clear that whenever target data are available, the optimally tuned pretraining-finetuning method is *always no worse than* the optiamlly tuned pretraining method, as one can simply set the finetuning stepsize to be small (or zero) so that the former reduces to the latter. Moreover, Example 4.3 shows a covariate shift problem instance such that pretraining-finetuning can save *polynomially* amount of data compared to pretraining (or supervised learning). This example demonstrates the limitation of pretraining and the benefits of finetuning. As a final remark for Example 4.3, direct computation implies that $\mathbb{K}^* = \{1, 2, \dots, 2\epsilon^{-0.5} + 1\}$ and $\mathbb{K}^\dagger = \{2, 3, \dots, 2\epsilon^{-0.5} + 1\}$, therefore $\|\mathbf{H}_{\mathbb{K}^*}\|_{\mathbf{G}} = \epsilon^{-2} \gg \|\mathbf{H}_{\mathbb{K}^\dagger}\|_{\mathbf{G}} = \epsilon^{0.5}$, so the implication of Example 4.3 is consistent with Theorems 4.1 and 4.2.

**Numerical Simulations.** We perform experiments on synthetic data to verify our theory. The code and data for our experiments can be found on Github [3]. Recall that the effectiveness of pretraining and finetuning depends on the alignment between source and target covariance matrices, therefore we design experiments where the source and target covariance matrices are aligned at different levels. In particular, we consider commutable matrices $\mathbf{G}$ and $\mathbf{H}$ with eigenvalues $\{\mu_i\}_{i\geq 1} = \{i^{-2}\}_{i\geq 1}$ and $(\lambda_i)_{i\geq 1} = (i^{-1.5})_{i\geq 1}$, respectively. To simulate different alignments between $\mathbf{G}$ and $\mathbf{H}$, we first sort them so that both of their eigenvalues are in descending order, and then reverse the top-$k$ eigenvalues

---

[3] https://github.com/uclaml/pretrain-finetune-SGD

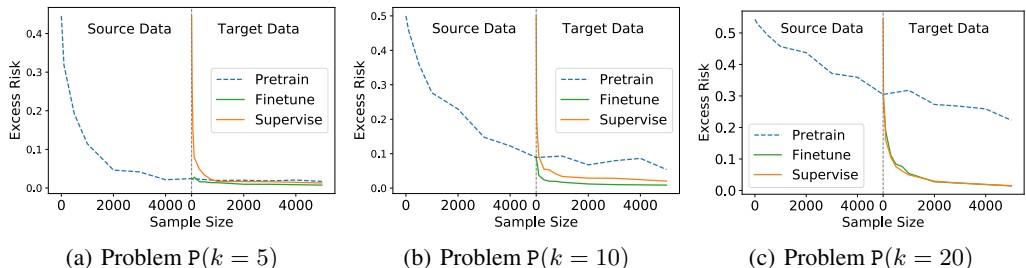

(a) Problem P($k = 5$)      (b) Problem P($k = 10$)      (c) Problem P($k = 20$)

Figure 1: A generalization comparison between pretraining, pretraining-finetuning, and supervised learning. For each point in the curves, its x-axis represents the sample size and its y-axis represents the excess risk achieved by an algorithm with the corresponding amount of samples under the optimally tuned stepsizes. For the pretraining curves, the sample size refers to the amount of source data and the sample size appeared in the right half of the plots should be added by $5,000$. The finetuning curves are generated from an initial model pretrained with $5,000$ source data and its sample size refers to the amount of target data. For supervised learning curves, the sample size refers to the amount of target data. The problem instances P($k = 5$), P($k = 10$), and P($k = 20$) are designed according to (4). The problem dimension is 200. The results are averaged over 20 independent repeats.

of $\mathbf{G}$. In mathematics, for a given $k > 0$, the problem instance P($k$) $:= (\mathbf{w}^*, \mathbf{H}, \mathbf{G}, \sigma^2)$ is designed as follows:

$$\mathbf{H} = \text{diag}\left(1, \frac{1}{2^{1.5}}, \dots, \frac{1}{k^{1.5}}, \frac{1}{(k+1)^{1.5}}, \dots\right), \quad \mathbf{G} = \text{diag}\left(\frac{1}{k^2}, \dots, \frac{1}{2^2}, 1, \frac{1}{(k+1)^2}, \dots\right),$$
$$\mathbf{w}^* = \big(\underbrace{1, 1, \dots, 1}_{k \text{ copies}}, 1/(k+1), 1/(k+2), \dots\big)^\top, \qquad \sigma^2 = 1. \tag{4}$$

One can verify that $\text{tr}(\mathbf{H}) \eqsim \text{tr}(\mathbf{G}) \eqsim 1$ and that $\|\mathbf{w}^*\|_\mathbf{H}^2 \eqsim \|\mathbf{w}^*\|_\mathbf{G}^2 \eqsim \sigma^2 \eqsim 1$. Clearly, a larger $k$ implies a worse alignment between $\mathbf{G}$ and $\mathbf{H}$. We then test three problem instances P($k = 5$), P($k = 10$), and P($k = 20$), and compare the excess risk achieved by pretraining, pretraining-finetuning, and supervised learning. The results are presented in Figure 1, which lead to the following informative observations:

- For problem P($k = 5$) where $\mathbf{H}$ and $\mathbf{G}$ are aligned very well, pretraining (without finetuning!) can already match the generalization performance of supervised learning. This verifies the power of pretraining for tackling transfer learning with mildly shifted covariate.
- For problem P($k = 10$) where $\mathbf{H}$ and $\mathbf{G}$ are moderately aligned, there is a significant gap between the risk of pretraining and that of supervised learning. Yet, the gap is closed when the pretrained model is finetuned with scarce target data. This demonstrates the limitation of pretraining and the power of finetuning for tackling transfer learning with moderate shifted covariate.
- For problem P($k = 20$) where $\mathbf{H}$ and $\mathbf{G}$ are poorly aligned, the risk of pretraining can hardly compete with that of supervised learning. Moreover, for finetuning to match the performance of supervised learning, it requires nearly the same amount of target data as that used by supervised learning. This reveals the limitation of pretraining and finetuning for tackling transfer learning with severely shifted covariate.

## 5 Concluding Remarks

We consider linear regression under covariate shift, and a SGD estimator that is firstly trained with source domain data and then finetuned with target domain data. We derive sharp upper and lower bounds for the estimator's target domain excess risk. Based on the derived bounds, we show that for a large class of covariate shift problems, pretraining with $O(N^2)$ source data can match the performance of supervised learning with $N$ target data. Moreover, we show that finetuning with scarce target data can significantly reduce the amount of source data required by pretraining. Finally, when applied to supervised linear regression, our results improve the upper bound in [Wu et al., 2021] by a logarithmic factor, and close its gap with the lower bound (ignoring constant factors) when the signal-to-noise ratio is bounded.

Several future directions are worth discussing.

**Model Shift.** An immediate follow-up problem is to extend our results from the covariate shift setting to more general transfer learning settings, e.g., with both covariate shift and *model shift*, where the true parameter $\mathbf{w}^*$ could also be different for source and target distributions. Under model shift, the power of pretraining with source data is limited, and we expect that finetuning with target data becomes even more important.

**Ridge Regression.** For infinite-dimensional least-squares in the supervised learning context, instance-wisely tight bounds for both ridge regression and SGD have been established by Bartlett et al. [2020], Tsigler and Bartlett [2020], Zou et al. [2021b], Wu et al. [2021]. For infinite-dimensional least-squares under covariate shift, this paper presents nearly instance-wisely tight bounds for SGD. As ridge regression is popular in covariate shift literature (see Ma et al. [2022] and references herein), an interesting future direction is studying the instance-wisely tight bounds for ridge regression in the setting of infinite-dimensional least-squares under covariate shift — a tight bias analysis is of particular interest.

**Unlabeled Data.** In this work we assume the provided source and target data are both labeled. However in many practical scenarios, additional unlabeled source and target data are also available. In this case our results cannot be directly applied as it remains unclear how to utilize unlabeled data with SGD. An important future direction is to extend our framework to incorporate with unlabeled source and target data.

## Acknowledgments and Disclosure of Funding

We would like to thank the anonymous reviewers and area chairs for their helpful comments. This work was done when DZ was a Ph.D. student at UCLA. DZ is partially supported by Bloomberg data science Ph.D. fellowship and his startup funding in the institute of data science, the University of Hong Kong. JW and VB are supported by the Defense Advanced Research Projects Agency (DARPA) under Contract No. HR00112190130. QG is partially supported by the National Science Foundation award IIS-1906169 and IIS-2008981. SK acknowledges funding from the Office of Naval Research under award N00014-22-1-2377 and the National Science Foundation Grant under award CCF-1703574. The views and conclusions contained in this paper are those of the authors and should not be interpreted as representing any funding agencies.

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
