_{\mathrm{eff}}^{\mathtt{sl}} \gamma^{\mathtt{sl}})\}, \quad D_{\mathrm{eff}}^{\mathtt{sl}} := |\mathbb{K}^*| + (N_{\mathrm{eff}}^{\mathtt{sl}} \gamma^{\mathtt{sl}})^2 \sum_{i \notin \mathbb{K}^*} \lambda_i^2,$$

*and $\gamma^{\mathtt{sl}} < 1/\|\mathbf{H}\|_2$ refers to the optimal initial stepsize for supervised learning.*

We now explain the implication of Theorem 4.1. First of all, it is of statistical interest to consider a signal-to-noise ratio bounded from above, i.e., $\|\mathbf{w}^*\|_{\mathbf{G}}^2 / \sigma^2 \lesssim 1$. Note that $D_{\mathrm{eff}}^{\mathtt{sl}} \geq 1$ when $N^{\mathtt{sl}}$ is large. Moreover, recall that $|\mathbb{K}^*| \leq D_{\mathrm{eff}}^{\mathtt{sl}} = o(N_{\mathrm{eff}}^{\mathtt{sl}})$ when supervised learning can achieve a vanishing excess risk. Finally, $\|\mathbf{H}_{\mathbb{K}^*}\|_{\mathbf{G}} := \|\mathbf{G}^{-\frac{1}{2}} \mathbf{H}_{\mathbb{K}^*} \mathbf{G}^{-\frac{1}{2}}\|_2 \lesssim 1$ can be satisfied if the top $|\mathbb{K}^*| = o(N_{\mathrm{eff}}^{\mathtt{sl}})$ eigenvalues subspace of $\mathbf{H}$ mostly falls into the top eigenvalues subspace of $\mathbf{G}$. Under these remarks, Theorem 4.1 suggests that: in the bounded signal-to-noise cases, pretraining with $O(N^2)$ source data is *no worse than* supervised learning with $N$ target data (ignoring constant factors), for *every* covariate shift problem such that the *top* eigenvalues subspace of the target covariance matrix aligns well with that of the source covariance matrix.

**The Power of Pretraining-Finetuning.** We next discuss the effect of pretraining-finetuning by comparing Theorem 3.1 with Corollary 3.4.

**Theorem 4.2** (Pretraining-finetuning vs. supervised learning). *Suppose that Assumptions 1 and 2' hold. Let $\mathbf{w}_{0+N^{\mathtt{sl}}}$ be the output of* (SGD) *with optimally tuned initial stepsize, 0 source data and $N^{\mathtt{sl}} > 100$ target data. Let $\mathbf{w}_{M+N}$ be the output of* (SGD) *with optimally tuned initial stepsize,*

$M$ source data and $N$ target data. Let $M_{\texttt{eff}} := M/\log(M)$, $N_{\texttt{eff}} := N/\log(N)$, $N_{\texttt{eff}}^{\texttt{sl}} := N^{\texttt{sl}}/\log(N^{\texttt{sl}})$. Suppose all SGD methods are initialized from $\mathbf{0}$. Then for every covariate shift problem instance $(\mathbf{w}^*, \mathbf{H}, \mathbf{G})$ such that $\mathbf{H}, \mathbf{G}$ commute, it holds that

$$\texttt{ExcessRisk}(\mathbf{w}_{M+N}) \lesssim (1 + \alpha \|\mathbf{w}^*\|_{\mathbf{G}}^2/\sigma^2) \cdot \texttt{ExcessRisk}(\mathbf{w}_{0+N^{\texttt{sl}}})$$

*provided that*

$$M_{\texttt{eff}} \geq (N_{\texttt{eff}}^{\texttt{sl}})^2 \cdot \frac{4\|\mathbf{H}_{\mathbb{K}^\dagger}\|_{\mathbf{G}}}{\alpha D_{\texttt{eff}}^{\texttt{sl}}},$$

*where*

$$\mathbb{K}^\dagger := \{k : N_{\texttt{eff}}^{\texttt{sl}} \log(N_{\texttt{eff}}^{\texttt{sl}}) \operatorname{tr}(\mathbf{H})/(N_{\texttt{eff}} D_{\texttt{eff}}^{\texttt{sl}}) > \lambda_k \geq 1/(N_{\texttt{eff}}^{\texttt{sl}} \gamma^{\texttt{sl}})\} \subset \mathbb{K}^*,$$

*and $\mathbb{K}^*$, $D_{\texttt{eff}}^{\texttt{sl}}$, $\gamma^{\texttt{sl}}$ are as defined in Theorem 4.1.*

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

of $\mathbf{G}$. In mathematics, for a given $k > 0$, the problem instance P$(k) := (\mathbf{w}^*, \mathbf{H}, \mathbf{G}, \sigma^2)$ is designed as follows:

$$\mathbf{H} = \mathrm{diag}\left(1, \frac{1}{2^{1.5}}, \ldots, \frac{1}{k^{1.5}}, \frac{1}{(k+1)^{1.5}}, \ldots\right), \quad \mathbf{G} = \mathrm{diag}\left(\frac{1}{k^2}, \ldots, \frac{1}{2^2}, 1, \frac{1}{(k+1)^2}, \ldots\right),$$

$$\mathbf{w}^* = \big(\underbrace{1, 1, \ldots, 1}_{k \text{ copies}}, 1/(k+1), 1/(k+2), \ldots\big)^\top, \qquad \sigma^2 = 1. \tag{4}$$

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

## Acknowledgments and Disclosure of Funding

We would like to thank the anonymous reviewers and area chairs for their helpful comments. This work was done when DZ was a Ph.D. student at UCLA. DZ is partially supported by Bloomberg data science Ph.D. fellowship and his startup funding in the institute of data science, the University of Hong Kong. JW and VB are supported by the Defense Advanced Research Projects Agency (DARPA) under Contract No. HR00112190130. QG is partially supported by the National Science Foundation award IIS-1906169 and IIS-2008981. SK acknowledges funding from the Office of Naval Research under award N00014-22-1-2377 and the National Science Foundation Grant under award CCF-1703574. The views and conclusions contained in this paper are those of the authors and should not be interpreted as representing any funding agencies.

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

# A   A Comparison of Pretraining-Finetuning, Pretraining and Supervised Learning

In Table 1, we make a detailed comparison of the bounds for (1) pretraining-finetuning with $M$ source data and $N$ target data, (2) pretraining with $M$ source data, and (3) supervised learning with $N$ target data. The presented bounds are from Theorem 3.1, Corollaries 3.3 and 3.4. For simplicity, we assume that all SGD iterates are initialized from $\mathbf{w}_0 = 0$, and that the signal-to-noise ratios are bounded from above.

Table 1:   A comparison of pretraining-finetuning, pretraining and supervised learning.

| | Pretraining-Finetuning | Pretraining | Supervised Learning |
|---|---|---|---|
| initial stepsizes | $\gamma_0,\ \gamma_M$ | $\gamma_0$ | $\gamma_0 (= \gamma_M)$ |
| number of data | $M + N$ | $M + 0$ | $0 + N$ |
| effective number of source data ($M_{\mathtt{eff}}$) | $\dfrac{M}{\log(M)}$ | $\dfrac{M}{\log(M)}$ | $0$ |
| effective number of target data ($N_{\mathtt{eff}}$) | $\dfrac{N}{\log(N)}$ | $0$ | $\dfrac{N}{\log(N)}$ |
| source effective dimension ($D_{\mathtt{eff}}^{\mathtt{source}}$) | $\mathrm{tr}\Big(\prod\limits_{t=0}^{N-1}(\mathbf{I}-\gamma_{M+t}\mathbf{H})^2\mathbf{H}\cdot \big(\mathbf{G}_{\mathbb{J}}^{-1}+M_{\mathtt{eff}}^2\gamma_0^2\mathbf{G}_{\mathbb{J}^c}\big)\Big)$ | $\mathrm{tr}(\mathbf{H}\mathbf{G}_{\mathbb{J}}^{-1})+ M_{\mathtt{eff}}^2\gamma_0^2\,\mathrm{tr}(\mathbf{H}\mathbf{G}_{\mathbb{J}^c})$ | $0$ |
| target effective dimension ($D_{\mathtt{eff}}^{\mathtt{target}}$) | $|\mathbb{K}| + N_{\mathtt{eff}}^2\gamma_M^2\sum_{i\notin\mathbb{K}}\lambda_i^2$ | $0$ | $|\mathbb{K}| + N_{\mathtt{eff}}^2\gamma_0^2\sum_{i\notin\mathbb{K}}\lambda_i^2$ |
| learnable indexes | $\mathbb{J} = \{j : \mu_j \ge 1/(\gamma_0 M_{\mathtt{eff}})\}, \quad \mathbb{K} = \{k : \lambda_k \ge 1/(\gamma_M N_{\mathtt{eff}})\}$ | | |
| Signal to noise ratio (SNR) | $\dfrac{\|\mathbf{w}^*\|_{\mathbf{G}}^2 + \|\prod_{t=0}^{M-1}(\mathbf{I}-\gamma_t\mathbf{G})\mathbf{w}^*\|_{\mathbf{H}}^2}{\sigma^2}$ | $\dfrac{\|\mathbf{w}^*\|_{\mathbf{G}}^2}{\sigma^2}$ | $\dfrac{\|\mathbf{w}^*\|_{\mathbf{H}}^2}{\sigma^2}$ |
| effective bias error ($\mathtt{bias}_{\mathtt{eff}}$) | $\Big\|\prod\limits_{t=M}^{M+N-1}(\mathbf{I}-\gamma_t\mathbf{H})\prod\limits_{t=0}^{M-1}(\mathbf{I}-\gamma_t\mathbf{G})\mathbf{w}^*\Big\|_{\mathbf{H}}^2$ | $\Big\|\prod\limits_{t=0}^{N-1}(\mathbf{I}-\gamma_t\mathbf{G})\mathbf{w}^*\Big\|_{\mathbf{H}}^2$ | $\Big\|\prod\limits_{t=0}^{N-1}(\mathbf{I}-\gamma_t\mathbf{H})\mathbf{w}^*\Big\|_{\mathbf{H}}^2$ |
| unified risk bound | $\mathtt{bias}_{\mathtt{eff}} + (1+\sigma^2)\cdot\mathtt{SNR}\cdot\left(\dfrac{D_{\mathtt{eff}}^{\mathtt{source}}}{M_{\mathtt{eff}}} + \dfrac{D_{\mathtt{eff}}^{\mathtt{target}}}{N_{\mathtt{eff}}}\right)$ | | |

# B   Preliminaries

**Notations.**   Define the following operators on symmetric matrices:

$$\mathcal{I} := \mathbf{I}\otimes\mathbf{I},$$
$$\mathcal{M}_{\mathbf{G}} := \mathbb{E}_{\mathtt{source}}[\mathbf{x}\otimes\mathbf{x}\otimes\mathbf{x}\otimes\mathbf{x}], \qquad \mathcal{M}_{\mathbf{H}} := \mathbb{E}_{\mathtt{target}}[\mathbf{x}\otimes\mathbf{x}\otimes\mathbf{x}\otimes\mathbf{x}],$$

$$\widetilde{\mathcal{M}}_{\mathbf{G}} := \mathbf{G} \otimes \mathbf{G}, \qquad \widetilde{\mathcal{M}}_{\mathbf{H}} := \mathbf{H} \otimes \mathbf{H},$$
$$\mathcal{T}_{\mathbf{G}}(\gamma) := \mathbf{G} \otimes \mathbf{I} + \mathbf{I} \otimes \mathbf{G} - \gamma \mathcal{M}_{\mathbf{G}}, \qquad \mathcal{T}_{\mathbf{H}}(\gamma) := \mathbf{H} \otimes \mathbf{I} + \mathbf{I} \otimes \mathbf{H} - \gamma \mathcal{M}_{\mathbf{H}},$$
$$\widetilde{\mathcal{T}}_{\mathbf{G}}(\gamma) := \mathbf{G} \otimes \mathbf{I} + \mathbf{I} \otimes \mathbf{G} - \gamma \mathbf{G} \otimes \mathbf{G}, \qquad \widetilde{\mathcal{T}}_{\mathbf{H}}(\gamma) := \mathbf{H} \otimes \mathbf{I} + \mathbf{I} \otimes \mathbf{H} - \gamma \mathbf{H} \otimes \mathbf{H}$$

For the linear operators we have the following technical lemma from Zou et al. [2021b].

**Lemma B.1** (Lemma B.1, Zou et al. [2021b]). *An operator $\mathcal{O}$ defined on symmetric matrices is called PSD mapping, if $\mathbf{A} \succeq 0$ implies $\mathcal{O} \circ \mathbf{A} \succeq 0$. Then we have*

1. $\mathcal{M}_{\mathbf{G}}$, $\mathcal{M}_{\mathbf{H}}$, $\widetilde{\mathcal{M}}_{\mathbf{G}}$ and $\widetilde{\mathcal{M}}_{\mathbf{H}}$ are all PSD mappings.

2. $\mathcal{I} - \gamma \mathcal{T}_{\mathbf{G}}(\gamma)$, $\mathcal{I} - \gamma \mathcal{T}_{\mathbf{H}}(\gamma)$, $\mathcal{I} - \gamma \widetilde{\mathcal{T}}_{\mathbf{G}}(\gamma)$ and $\mathcal{I} - \gamma \widetilde{\mathcal{T}}_{\mathbf{H}}(\gamma)$ are all PSD mappings.

3. $\mathcal{M}_{\mathbf{G}} - \widetilde{\mathcal{M}}_{\mathbf{G}}$, $\mathcal{M}_{\mathbf{H}} - \widetilde{\mathcal{M}}_{\mathbf{H}}$, $\widetilde{\mathcal{T}}_{\mathbf{G}} - \mathcal{T}_{\mathbf{G}}$ and $\widetilde{\mathcal{T}}_{\mathbf{H}} - \mathcal{T}_{\mathbf{H}}$ are all PSD mappings.

4. *If $0 < \gamma < 1/\|\mathbf{G}\|_2$, then $\widetilde{\mathcal{T}}_{\mathbf{G}}^{-1}$ exists, and is a PSD mapping. Similarly, if $0 < \gamma < 1/\|\mathbf{H}\|_2$, then $\widetilde{\mathcal{T}}_{\mathbf{H}}^{-1}$ exists, and is a PSD mapping.*

5. *If $0 < \gamma < 1/(\alpha \operatorname{tr}(\mathbf{G}))$, then $\mathcal{T}_{\mathbf{G}}^{-1} \circ \mathbf{A}$ exists for PSD matrix $\mathbf{A}$, and $\mathcal{T}_{\mathbf{G}}^{-1}$ is a PSD mapping. Similarly, if $0 < \gamma < 1/(\alpha \operatorname{tr}(\mathbf{H}))$, then $\mathcal{T}_{\mathbf{H}}^{-1} \circ \mathbf{A}$ exists for PSD matrix $\mathbf{A}$, and $\mathcal{T}_{\mathbf{H}}^{-1}$ is a PSD mapping.*

6. *For every $\gamma > 0$ and every PSD matrices $\mathbf{A}$ and $\mathbf{B}$, we have*
$$\langle \mathbf{A}, (\mathcal{I} - \gamma \mathcal{T}_{\mathbf{G}}(\gamma)) \circ \mathbf{B} \rangle = \langle (\mathcal{I} - \gamma \mathcal{T}_{\mathbf{G}}(\gamma)) \circ \mathbf{A}, \mathbf{B} \rangle,$$
$$\langle \mathbf{A}, (\mathcal{I} - \gamma \mathcal{T}_{\mathbf{H}}(\gamma)) \circ \mathbf{B} \rangle = \langle (\mathcal{I} - \gamma \mathcal{T}_{\mathbf{H}}(\gamma)) \circ \mathbf{A}, \mathbf{B} \rangle.$$

*Proof.* Proof to the first five claims can be found in Lemma B.1 in Zou et al. [2021b]. The last claim is by definition. $\square$

Define
$$\boldsymbol{\Sigma}_{\mathbf{G}} := \mathbb{E}_{\texttt{source}}[(y - \langle \mathbf{w}^*, \mathbf{x} \rangle)^2 \mathbf{x}\mathbf{x}^\top], \quad \boldsymbol{\Sigma}_{\mathbf{H}} := \mathbb{E}_{\texttt{source}}[(y - \langle \mathbf{w}^*, \mathbf{x} \rangle)^2 \mathbf{x}.\mathbf{x}^\top]$$

Then for the SGD iterates, we can consider their associated bias iterates and variance iterates:

$$\begin{cases} \mathbf{B}_0 = (\mathbf{w}_0 - \mathbf{w}^*)(\mathbf{w}_0 - \mathbf{w}^*)^\top; \\ \mathbf{B}_t = (\mathcal{I} - \gamma_{t-1}\mathcal{T}_{\mathbf{G}}(\gamma_{t-1})) \circ \mathbf{B}_{t-1}, & t = 1, \dots, M; \\ \mathbf{B}_{M+t} = (\mathcal{I} - \gamma_{M+t-1}\mathcal{T}_{\mathbf{H}}(\gamma_{M+t-1})) \circ \mathbf{B}_{M+t-1}, & t = 1, \dots, N; \end{cases} \tag{5}$$

$$\begin{cases} \mathbf{C}_0 = \mathbf{0}; \\ \mathbf{C}_t = (\mathcal{I} - \gamma_{t-1}\mathcal{T}_{\mathbf{G}}(\gamma_{t-1})) \circ \mathbf{C}_{t-1} + \gamma_t^2 \boldsymbol{\Sigma}_{\mathbf{G}}, & t = 1, \dots, M; \\ \mathbf{C}_{M+t} = (\mathcal{I} - \gamma_{M+t-1}\mathcal{T}_{\mathbf{H}}(\gamma_{M+t-1})) \circ \mathbf{C}_{M+t-1} + \gamma_{M+t-1}^2 \boldsymbol{\Sigma}_{\mathbf{H}}, & t = 1, \dots, N. \end{cases} \tag{6}$$

**Lemma B.2** (Bias-variance decomposition). *Suppose that Assumption 2 holds. Then we have*
$$\mathbb{E}[\texttt{ExcessRisk}(\mathbf{w}_{M+N})] \leq \langle \mathbf{H}, \mathbf{B}_{M+N} \rangle + \langle \mathbf{H}, \mathbf{C}_{M+N} \rangle.$$

*Proof.* This follows from Lemma 2 in Wu et al. [2021]. $\square$

**Lemma B.3** (Bias-variance decomposition, lower bound). *Suppose that Assumption 2' holds. Then we have*
$$\mathbb{E}[\texttt{ExcessRisk}(\mathbf{w}_{M+N})] = \frac{1}{2}\langle \mathbf{H}, \mathbf{B}_{M+N} \rangle + \frac{1}{2}\langle \mathbf{H}, \mathbf{C}_{M+N} \rangle.$$

*Proof.* This follows from Lemma 3 in Wu et al. [2021]. $\square$

## C  Variance Error Analysis

### C.1  Upper Bounds

The following Assumption 1' is implied by Assumption 1A by setting $R^2 = \max\{\alpha\operatorname{tr}(\mathbf{H}), \alpha\operatorname{tr}(\mathbf{G})\}$. In this part we will work with the weaker Assumption 1'.

**Assumption 1'** (Fourth moment condition, relaxed version). *There exists a constant $R > 0$ such that*

$$\mathbb{E}_{\mathtt{source}}[\mathbf{x}\mathbf{x}^\top\mathbf{x}\mathbf{x}^\top] \preceq R^2\mathbf{G}, \quad \mathbb{E}_{\mathtt{target}}[\mathbf{x}\mathbf{x}^\top\mathbf{x}\mathbf{x}^\top] \preceq R^2\mathbf{H}.$$

**Lemma C.1** (Crude bound on the variance iterates). *Suppose that Assumptions 1' and 2 hold. Suppose that $\max\{\gamma_0, \gamma_M\} \leq \gamma < 1/R^2$. Then it holds that*

$$\mathbf{C}_t \leq \frac{\gamma\sigma^2}{1 - \gamma R^2}\mathbf{I}, \ \ \textit{for every } t = 0, 1, \ldots, M + N.$$

*Proof.* The proof idea has appeared in Jain et al. [2017a], Ge et al. [2019], Wu et al. [2021]. We prove the lemma by induction. For $t = 0$, it is clear that $\mathbf{C}_0 = \mathbf{0} \preceq \frac{\gamma\sigma^2}{1-\gamma R^2}\mathbf{I}$. Now suppose that $\mathbf{C}_{t-1} \leq \frac{\gamma\sigma^2}{1-\gamma R^2}\mathbf{I}$, and consider $\mathbf{C}_t$ according to (6). If $t \leq M$, then according to (6) we have

$$\begin{aligned}
\mathbf{C}_t &= \big(\mathcal{I} - \gamma_{t-1}\mathcal{T}_{\mathbf{G}}(\gamma_{t-1})\big) \circ \mathbf{C}_{t-1} + \gamma_{t-1}^2\mathbf{\Sigma}_{\mathbf{G}} \\
&\preceq \frac{\gamma\sigma^2}{1 - \gamma R^2} \cdot \big(\mathcal{I} - \gamma_{t-1}\mathcal{T}_{\mathbf{G}}(\gamma_{t-1})\big) \circ \mathbf{I} + \gamma_{t-1}^2\sigma^2 \cdot \mathbf{G} \qquad (7) \\
&\preceq \frac{\gamma\sigma^2}{1 - \gamma R^2} \cdot (\mathbf{I} - 2\gamma_{t-1}\mathbf{G} + \gamma_{t-1}^2 \cdot R^2 \cdot \mathbf{G}) + \gamma_{t-1}^2\sigma^2 \cdot \mathbf{G} \\
&= \frac{\gamma\sigma^2}{1 - \gamma R^2} \cdot \mathbf{I} - (2\gamma_{t-1}\gamma - \gamma_{t-1}^2) \cdot \frac{\sigma^2}{1 - \gamma R^2} \cdot \mathbf{G} \\
&\preceq \frac{\gamma\sigma^2}{1 - \gamma R^2} \cdot \mathbf{I}.
\end{aligned}$$

If $t > M$, similarly according to (6) we have

$$\begin{aligned}
\mathbf{C}_t &= \big(\mathcal{I} - \gamma_{t-1}\mathcal{T}_{\mathbf{H}}(\gamma_{t-1})\big) \circ \mathbf{C}_{t-1} + \gamma_{t-1}^2\mathbf{\Sigma}_{\mathbf{H}} \\
&\preceq \frac{\gamma\sigma^2}{1 - \gamma R^2} \cdot \big(\mathcal{I} - \gamma_{t-1}\mathcal{T}_{\mathbf{H}}(\gamma_{t-1})\big) \circ \mathbf{I} + \gamma_{t-1}^2\sigma^2 \cdot \mathbf{H} \qquad (8) \\
&\preceq \frac{\gamma\sigma^2}{1 - \gamma R^2} \cdot (\mathbf{I} - 2\gamma_{t-1}\mathbf{H} + \gamma_{t-1}^2 \cdot R^2 \cdot \mathbf{H}) + \gamma_{t-1}^2\sigma^2 \cdot \mathbf{H} \\
&= \frac{\gamma\sigma^2}{1 - \gamma R^2} \cdot \mathbf{I} - (2\gamma_{t-1}\gamma - \gamma_{t-1}^2) \cdot \frac{\sigma^2}{1 - \gamma R^2}\mathbf{H} \\
&\preceq \frac{\gamma\sigma^2}{1 - \gamma R^2} \cdot \mathbf{I}.
\end{aligned}$$

Putting everything together we complete the induction. $\qquad\square$

**Lemma C.2** (Upper bounds on the variance iterates). *Suppose that Assumptions 1' and 2 hold. Suppose that $\max\{\gamma_0, \gamma_M\} \leq \gamma < 1/R^2$. Let $M_{\mathtt{eff}} := M/\log(M)$, $N_{\mathtt{eff}} := N/\log(N)$.*

- *For every index set $\mathbb{J} \subset \mathbb{N}_+$, it holds that*

$$\mathbf{C}_M \preceq \frac{8\sigma^2}{1 - \gamma R^2} \cdot \Big(\frac{1}{M_{\mathtt{eff}}} \cdot \mathbf{G}_{\mathbb{J}}^{-1} + M_{\mathtt{eff}}\gamma_0^2 \cdot \mathbf{G}_{\mathbb{J}^c}\Big).$$

- *For every index set $\mathbb{K} \subset \mathbb{N}_+$, it holds that*

$$\sum_{t=0}^{N-1} \gamma_{M+t}^2 \prod_{i=t+1}^{N-1} (\mathbf{I} - \gamma_{M+i}\mathbf{H})^2\mathbf{H} \preceq 8 \cdot \Big(\frac{1}{N_{\mathtt{eff}}}\mathbf{H}_{\mathbb{K}}^{-1} + N_{\mathtt{eff}}\gamma_M^2 \cdot \mathbf{H}_{\mathbb{K}^c}\Big).$$

*Proof.* These are from the proof of Theorem 5 in Wu et al. [2021]. □

**Theorem C.1** (Variance error upper bound)**.** *Suppose that Assumptions 1' and 2 hold. Suppose that* $\max\{\gamma_0, \gamma_M\} \leq \gamma < 1/R^2$. *Let* $M_{\tt eff} := M/\log(M)$, $N_{\tt eff} := N/\log(N)$. *Then it holds that*

$$\langle \mathbf{H}, \mathbf{C}_{M+N} \rangle \leq \frac{8\sigma^2}{1-\gamma R^2} \cdot \left( \frac{D_{\tt eff}^{\tt finetune}}{M_{\tt eff}} + \frac{D_{\tt eff}}{N_{\tt eff}} \right),$$

*where*

$$D_{\tt eff} := \text{tr}(\mathbf{H}\mathbf{H}_{\mathbb{K}}^{-1}) + N_{\tt eff}^2 \gamma_M^2 \cdot \text{tr}(\mathbf{H}\mathbf{H}_{\mathbb{K}^c}),$$

$$D_{\tt eff}^{\tt finetune} := \text{tr}\left( \prod_{t=0}^{N-1} (\mathbf{I} - \gamma_{M+t}\mathbf{H})^2 \mathbf{H} \cdot \left( \mathbf{G}_{\mathbb{J}}^{-1} + M_{\tt eff}^2 \gamma_0^2 \cdot \mathbf{G}_{\mathbb{J}^c} \right) \right),$$

*and* $\mathbb{K}$, $\mathbb{J}$ *can be arbitrary index sets.*

*Proof of Theorem C.1.* The core idea is to relate $\mathbf{C}_{M+N}$ to $\mathbf{C}_M$ via (6). For every $t = 0, \ldots, N-1$, according to (6) we have

$$\begin{aligned}
\mathbf{C}_{M+t+1} &= \left( \mathcal{I} - \gamma_{M+t} \mathcal{T}_{\mathbf{H}}(\gamma_{M+t}) \right) \circ \mathbf{C}_{M+t} + \gamma_{M+t}^2 \mathbf{\Sigma}_{\mathbf{H}} \\
&\preceq \left( \mathcal{I} - \gamma_{M+t} \widetilde{\mathcal{T}}_{\mathbf{H}}(\gamma_{M+t}) \right) \circ \mathbf{C}_{M+t} + \gamma_{M+t}^2 \cdot \mathcal{M}_{\mathbf{H}} \circ \mathbf{C}_{M+t} + \gamma_{M+t}^2 \sigma^2 \cdot \mathbf{H} \\
&\preceq \left( \mathcal{I} - \gamma_{M+t} \widetilde{\mathcal{T}}_{\mathbf{H}}(\gamma_{M+t}) \right) \circ \mathbf{C}_{M+t} + \frac{\gamma_{M+t}^2 R^2 \cdot \gamma\sigma^2}{1-\gamma R^2} \cdot \mathbf{H} + \gamma_{M+t}^2 \sigma^2 \cdot \mathbf{H} \quad \text{(by Lemma C.1)} \\
&= \left( \mathcal{I} - \gamma_{M+t} \widetilde{\mathcal{T}}_{\mathbf{H}}(\gamma_{M+t}) \right) \circ \mathbf{C}_{M+t} + \frac{\gamma_{M+t}^2 \sigma^2}{1-\gamma R^2} \cdot \mathbf{H}.
\end{aligned}$$

Solving the above recursion from $t = 0$ to $t = N-1$ we obtain

$$\mathbf{C}_{M+N} \preceq \prod_{t=0}^{N-1} \left( \mathcal{I} - \gamma_{M+t} \widetilde{\mathcal{T}}_{\mathbf{H}}(\gamma_{M+t}) \right) \circ \mathbf{C}_M \qquad (9)$$

$$+ \frac{\sigma^2}{1-\gamma R^2} \cdot \sum_{t=0}^{N-1} \gamma_{M+t}^2 \prod_{i=t+1}^{N-1} \left( \mathcal{I} - \gamma_{M+i} \widetilde{\mathcal{T}}_{\mathbf{H}}(\gamma_{M+i}) \right) \circ \mathbf{H}$$

$$= \prod_{t=0}^{N-1} \left( \mathcal{I} - \gamma_{M+t} \widetilde{\mathcal{T}}_{\mathbf{H}}(\gamma_{M+t}) \right) \circ \mathbf{C}_M + \frac{\sigma^2}{1-\gamma R^2} \cdot \sum_{t=0}^{N-1} \gamma_{M+t}^2 \prod_{i=t+1}^{N-1} (\mathbf{I} - \gamma_{M+i}\mathbf{H})^2 \mathbf{H}.$$

Therefore the variance error is

$$\langle \mathbf{H}, \mathbf{C}_{M+N} \rangle$$

$$\leq \left\langle \mathbf{H}, \prod_{t=0}^{N-1} \left( \mathcal{I} - \gamma_{M+t} \widetilde{\mathcal{T}}_{\mathbf{H}}(\gamma_{M+t}) \right) \circ \mathbf{C}_M \right\rangle + \frac{\sigma^2}{1-\gamma R^2} \left\langle \mathbf{H}, \sum_{t=0}^{N-1} \gamma_{M+t}^2 \prod_{i=t+1}^{N-1} (\mathbf{I} - \gamma_{M+i}\mathbf{H})^2 \mathbf{H} \right\rangle$$

$$= \left\langle \prod_{t=0}^{N-1} \left( \mathcal{I} - \gamma_{M+t} \widetilde{\mathcal{T}}_{\mathbf{H}}(\gamma_{M+t}) \right) \circ \mathbf{H}, \mathbf{C}_M \right\rangle + \frac{\sigma^2}{1-\gamma R^2} \left\langle \mathbf{H}, \sum_{t=0}^{N-1} \gamma_{M+t}^2 \prod_{i=t+1}^{N-1} (\mathbf{I} - \gamma_{M+i}\mathbf{H})^2 \mathbf{H} \right\rangle$$

$$= \left\langle \prod_{t=0}^{N-1} (\mathbf{I} - \gamma_{M+t}\mathbf{H})^2 \mathbf{H}, \mathbf{C}_M \right\rangle + \frac{\sigma^2}{1-\gamma R^2} \left\langle \mathbf{H}, \sum_{t=0}^{N-1} \gamma_{M+t}^2 \prod_{i=t+1}^{N-1} (\mathbf{I} - \gamma_{M+i}\mathbf{H})^2 \mathbf{H} \right\rangle.$$

Finally, applying Lemma C.2 completes the proof. □

## C.2 Lower Bounds

**Lemma C.3** (Lower bounds on the variance iterates)**.** *Suppose that Assumptions 1B and 2' hold. Suppose that* $\gamma_0 < 1/\|\mathbf{G}\|_2$, $\gamma_M < 1/\|\mathbf{H}\|_2$. *Let* $M_{\tt eff} := M/\log(M)$, $N_{\tt eff} := N/\log(N)$.

- *For* $\mathbb{J} := \{j \in \mathbb{N}_+ : \mu_j \geq 1/(M_{\tt eff}\gamma_0)\}$, *it holds that*

$$\mathbf{C}_M \succeq \frac{\sigma^2}{400} \cdot \left( \frac{1}{M_{\tt eff}} \cdot \mathbf{G}_{\mathbb{J}}^{-1} + M_{\tt eff}\gamma_0^2 \cdot \mathbf{G}_{\mathbb{J}^c} \right).$$

- *For $\mathbb{K} := \{k \in \mathbb{N}_+ : \lambda_k \geq 1/(N_{\texttt{eff}}\gamma_M)\}$, it holds that*

$$\sum_{t=0}^{N-1} \gamma_{M+t}^2 \prod_{i=t+1}^{N-1} (\mathbf{I} - \gamma_{M+i}\mathbf{H})^2 \mathbf{H} \succeq \frac{1}{400} \cdot \left( \frac{1}{N_{\texttt{eff}}} \mathbf{H}_{\mathbb{K}}^{-1} + N_{\texttt{eff}}\gamma_M^2 \cdot \mathbf{H}_{\mathbb{K}^c} \right).$$

*Proof.* There are from the proof of Theorem 7 in Wu et al. [2021]. $\qquad\square$

**Theorem C.2** (Variance error lower bound ). *Suppose that Assumptions 1B and 2' hold. Suppose that $\gamma_0 < 1/\|\mathbf{G}\|_2$, $\gamma_M < 1/\|\mathbf{H}\|_2$. Let $M_{\texttt{eff}} := M/\log(M)$, $N_{\texttt{eff}} := N/\log(N)$. The it holds that*

$$\langle \mathbf{H}, \, \mathbf{C}_{M+N} \rangle \geq \frac{\sigma^2}{400} \cdot \left( \frac{D_{\texttt{eff}}^{\texttt{finetune}}}{M_{\texttt{eff}}} + \frac{D_{\texttt{eff}}}{N_{\texttt{eff}}} \right),$$

*where*

$$D_{\texttt{eff}} := \mathrm{tr}(\mathbf{H}\mathbf{H}_{\mathbb{K}}^{-1}) + N_{\texttt{eff}}^2 \gamma_M^2 \cdot \mathrm{tr}(\mathbf{H}\mathbf{H}_{\mathbb{K}^c}),$$

$$D_{\texttt{eff}}^{\texttt{finetune}} := \mathrm{tr}\left( \prod_{t=0}^{N-1} (\mathbf{I} - \gamma_{M+t}\mathbf{H})^2 \mathbf{H} \cdot \left( \mathbf{G}_{\mathbb{J}}^{-1} + M_{\texttt{eff}}^2 \gamma_0^2 \cdot \mathbf{G}_{\mathbb{J}^c} \right) \right),$$

*and*

$$\mathbb{K} := \{k \in \mathbb{N}_+ : \lambda_k \geq 1/(N_{\texttt{eff}}\gamma_M)\}, \quad \mathbb{J} := \{j \in \mathbb{N}_+ : \mu_j \geq 1/(M_{\texttt{eff}}\gamma_0)\}.$$

*Proof of Theorem C.2.* The proof idea is similar to that of Theorem C.1. For every $t = 0, \dots, N-1$, it holds that

$$\mathbf{C}_{M+t+1} = \left( \mathcal{I} - \gamma_{M+t}\mathcal{T}_{\mathbf{H}}(\gamma_{M+t}) \right) \circ \mathbf{C}_{M+t} + \gamma_{M+t}^2 \sigma^2 \cdot \mathbf{H}$$
$$\succeq \left( \mathcal{I} - \gamma_{M+t}\widetilde{\mathcal{T}}_{\mathbf{H}}(\gamma_{M+t}) \right) \circ \mathbf{C}_{M+t} + \gamma_{M+t}^2 \sigma^2 \cdot \mathbf{H}.$$

Solving the above recursion from $t = 0$ to $t = N-1$ we obtain

$$\mathbf{C}_{M+N} \succeq \prod_{t=0}^{N-1} \left( \mathcal{I} - \gamma_{M+t}\widetilde{\mathcal{T}}_{\mathbf{H}}(\gamma_{M+t}) \right) \circ \mathbf{C}_M + \sigma^2 \cdot \sum_{t=0}^{N-1} \gamma_{M+t}^2 \prod_{i=t+1}^{N-1} \left( \mathcal{I} - \gamma_{M+i}\widetilde{\mathcal{T}}_{\mathbf{H}}(\gamma_{M+i}) \right) \circ \mathbf{H}$$
$$= \prod_{t=0}^{N-1} \left( \mathcal{I} - \gamma_{M+t}\widetilde{\mathcal{T}}_{\mathbf{H}}(\gamma_{M+t}) \right) \circ \mathbf{C}_M + \sigma^2 \cdot \sum_{t=0}^{N-1} \gamma_{M+t}^2 \prod_{i=t+1}^{N-1} (\mathbf{I} - \gamma_{M+i}\mathbf{H})^2 \mathbf{H}.$$

Therefore the variance error is

$$\langle \mathbf{H}, \, \mathbf{C}_{M+N} \rangle$$
$$\geq \left\langle \mathbf{H}, \, \prod_{t=0}^{N-1} \left( \mathcal{I} - \gamma_{M+t}\widetilde{\mathcal{T}}_{\mathbf{H}}(\gamma_{M+t}) \right) \circ \mathbf{C}_M \right\rangle + \sigma^2 \left\langle \mathbf{H}, \, \sum_{t=0}^{N-1} \gamma_{M+t}^2 \prod_{i=t+1}^{N-1} (\mathbf{I} - \gamma_{M+i}\mathbf{H})^2 \mathbf{H} \right\rangle$$
$$= \left\langle \prod_{t=0}^{N-1} \left( \mathcal{I} - \gamma_{M+t}\widetilde{\mathcal{T}}_{\mathbf{H}}(\gamma_{M+t}) \right) \circ \mathbf{H}, \, \mathbf{C}_M \right\rangle + \sigma^2 \left\langle \mathbf{H}, \, \sum_{t=0}^{N-1} \gamma_{M+t}^2 \prod_{i=t+1}^{N-1} (\mathbf{I} - \gamma_{M+i}\mathbf{H})^2 \mathbf{H} \right\rangle$$
$$= \left\langle \prod_{t=0}^{N-1} (\mathbf{I} - \gamma_{M+t}\mathbf{H})^2 \mathbf{H}, \, \mathbf{C}_M \right\rangle + \sigma^2 \left\langle \mathbf{H}, \, \sum_{t=0}^{N-1} \gamma_{M+t}^2 \prod_{i=t+1}^{N-1} (\mathbf{I} - \gamma_{M+i}\mathbf{H})^2 \mathbf{H} \right\rangle.$$

Finally, applying Lemma C.3 completes the proof. $\qquad\square$

## D  Bias Error Analysis

### D.1  Upper Bounds

**Lemma D.1** (Bounds on the summation of bias iterates). *Suppose that Assumption 1A holds. Suppose that $\gamma < 1/(\alpha \, \mathrm{tr}(\mathbf{G}))$. Then for every $n \geq 1$, it holds that*

$$\frac{1}{2\gamma} \cdot \left( \mathbf{I} - (\mathbf{I} - \gamma\mathbf{G})^{2n} \right) \preceq \sum_{t=0}^{n-1} \left( \mathcal{I} - \gamma \cdot \mathcal{T}_{\mathbf{G}}(\gamma) \right)^t \circ \mathbf{G} \preceq \frac{1}{\gamma} \cdot \left( \mathbf{I} - (\mathbf{I} - \gamma\mathbf{G})^{2n} \right).$$

*Proof.* By definition and Assumption 1A, we have

$$\mathcal{T}_{\mathbf{G}}(\gamma) \circ \mathbf{I} = 2\mathbf{G} - \gamma \cdot \mathcal{M}_{\mathbf{G}} \circ \mathbf{I} \begin{cases} \preceq 2\mathbf{G}; \\ \succeq \mathbf{G}. \end{cases}$$

Notice that $\mathcal{T}_{\mathbf{G}}^{-1}(\gamma)$ is a PSD mapping (when operates on PSD matrices), therefore

$$\frac{1}{2} \cdot \mathbf{I} \preceq \mathcal{T}_{\mathbf{G}}^{-1}(\gamma) \circ \mathbf{G} \preceq \mathbf{I}.$$

Similarly, $\widetilde{\mathcal{T}}_{\mathbf{G}}^{-1}(\gamma)$ is also a PSD mapping and that we have

$$\widetilde{\mathcal{T}}_{\mathbf{G}}(\gamma) \circ \mathbf{I} = 2\mathbf{G} - \gamma \cdot \mathbf{G}^2 \begin{cases} \preceq 2\mathbf{G}; \\ \succeq \mathbf{G}, \end{cases}$$

therefore

$$\frac{1}{2} \cdot \mathbf{I} \preceq \widetilde{\mathcal{T}}_{\mathbf{G}}^{-1}(\gamma) \circ \mathbf{G} \preceq \mathbf{I}.$$

With the above, we prove the upper bound as follows:

$$\sum_{t=0}^{n-1} \left(\mathcal{I} - \gamma \cdot \mathcal{T}_{\mathbf{G}}(\gamma)\right)^t \circ \mathbf{G} = \frac{1}{\gamma} \cdot \left(\mathcal{I} - \left(\mathcal{I} - \gamma\mathcal{T}_{\mathbf{G}}(\gamma)\right)^n\right) \circ \mathcal{T}_{\mathbf{G}}^{-1}(\gamma) \circ \mathbf{G}$$

$$\preceq \frac{1}{\gamma} \cdot \left(\mathcal{I} - \left(\mathcal{I} - \gamma\widetilde{\mathcal{T}}_{\mathbf{G}}(\gamma)\right)^n\right) \circ \mathcal{T}_{\mathbf{G}}^{-1}(\gamma) \circ \mathbf{G} \quad \text{(since } \widetilde{\mathcal{T}} - \mathcal{T} \text{ is PSD)}$$

$$\preceq \frac{1}{\gamma} \cdot \left(\mathcal{I} - \left(\mathcal{I} - \gamma\widetilde{\mathcal{T}}_{\mathbf{G}}(\gamma)\right)^n\right) \circ \mathbf{I} \quad \text{(since } \mathcal{T}_{\mathbf{G}}^{-1}(\gamma) \circ \mathbf{G} \preceq \mathbf{I}\text{)}$$

$$= \frac{1}{\gamma} \cdot \left(\mathbf{I} - (\mathbf{I} - \gamma\mathbf{G})^{2n}\right).$$

The lower bound is because

$$\sum_{t=0}^{n-1} \left(\mathcal{I} - \gamma \cdot \mathcal{T}_{\mathbf{G}}(\gamma)\right)^t \circ \mathbf{G} \succeq \sum_{t=0}^{n-1} \left(\mathcal{I} - \gamma \cdot \widetilde{\mathcal{T}}_{\mathbf{G}}(\gamma)\right)^t \circ \mathbf{G} \quad \text{(since } \widetilde{\mathcal{T}} - \mathcal{T} \text{ is PSD )}$$

$$= \frac{1}{\gamma} \cdot \left(\mathcal{I} - \left(\mathcal{I} - \gamma\widetilde{\mathcal{T}}_{\mathbf{G}}(\gamma)\right)^T\right) \circ \widetilde{\mathcal{T}}_{\mathbf{G}}^{-1}(\gamma) \circ \mathbf{G}$$

$$\succeq \frac{1}{2\gamma} \cdot \left(\mathcal{I} - \left(\mathcal{I} - \gamma\widetilde{\mathcal{T}}_{\mathbf{G}}(\gamma)\right)^T\right) \circ \mathbf{I} \quad \text{(since } \widetilde{\mathcal{T}}_{\mathbf{G}}^{-1}(\gamma) \circ \mathbf{G} \succeq 0.5\mathbf{I}\text{)}$$

$$= \frac{1}{2\gamma} \cdot \left(\mathbf{I} - (\mathbf{I} - \gamma\mathbf{G})^{2n}\right).$$

$\square$

**Lemma D.2** (Crude bounds on the bias iterates). *Suppose that Assumption 1A holds. Suppose that $\gamma < 1/(2\alpha \operatorname{tr}(\mathbf{G}))$. Then the following holds for every $n \geq 0$:*

$$\left(\mathcal{I} - \gamma \cdot \mathcal{T}_{\mathbf{G}}(\gamma)\right)^n \circ \mathbf{G} \preceq \begin{cases} (1 + \alpha\gamma\operatorname{tr}(\mathbf{G})) \cdot \mathbf{G}, \\ \frac{1}{1 - 2\alpha\gamma\operatorname{tr}(\mathbf{G})} \cdot \frac{1}{\max\{n,1\}\gamma} \cdot \mathbf{I}. \end{cases}$$

*In particular, the following holds for every $n \geq 1$ and every index set $\mathbb{J} \subset \mathbb{N}_+$:*

$$\left(\mathcal{I} - \gamma \cdot \mathcal{T}_{\mathbf{G}}(\gamma)\right)^n \circ \mathbf{G} \preceq \frac{1}{1 - 2\alpha\gamma\operatorname{tr}(\mathbf{G})} \cdot \left(\frac{\mathbf{I}_{\mathbb{J}}}{n\gamma} + \mathbf{G}_{\mathbb{J}^c}\right).$$

*Proof.* Notice the following decomposition:

$$\left(\mathcal{I} - \gamma\mathcal{T}_{\mathbf{G}}(\gamma)\right)^n \circ \mathbf{G}$$

$$= \left(\mathcal{I} - \gamma\widetilde{\mathcal{T}}_{\mathbf{G}}(\gamma)\right)^n \circ \mathbf{G} + \gamma^2 \sum_{t=0}^{n-1} \left(\mathcal{I} - \gamma\widetilde{\mathcal{T}}_{\mathbf{G}}(\gamma)\right)^{n-1-t} \circ (\mathcal{M}_{\mathbf{G}} - \widetilde{\mathcal{M}}_{\mathbf{G}}) \circ \left(\mathcal{I} - \gamma\mathcal{T}_{\mathbf{G}}(\gamma)\right)^t \circ \mathbf{G}$$

$$\preceq (\mathbf{I} - \gamma\mathbf{G})^{2n}\mathbf{G} + \alpha\gamma^2 \sum_{t=0}^{n-1} \left(\mathcal{I} - \gamma\widetilde{\mathcal{T}}_{\mathbf{G}}(\gamma)\right)^{n-1-t} \circ \mathbf{G} \cdot \left\langle \mathbf{G}, \left(\mathcal{I} - \gamma\mathcal{T}_{\mathbf{G}}(\gamma)\right)^t \circ \mathbf{G} \right\rangle$$

$$= (\mathbf{I} - \gamma\mathbf{G})^{2n}\mathbf{G} + \alpha\gamma^2 \sum_{t=0}^{n-1} (\mathbf{I} - \gamma\mathbf{G})^{2(n-1-t)}\mathbf{G} \cdot \left\langle \mathbf{G}, \left(\mathcal{I} - \gamma\mathcal{T}_{\mathbf{G}}(\gamma)\right)^t \circ \mathbf{G} \right\rangle. \tag{10}$$

Based on (10) we show the first conclusion as follows:

$$\left(\mathcal{I} - \gamma\mathcal{T}_{\mathbf{G}}(\gamma)\right)^n \circ \mathbf{G} \preceq \mathbf{G} + \alpha\gamma^2 \sum_{t=0}^{n-1} \mathbf{G} \cdot \left\langle \mathbf{G}, \left(\mathcal{I} - \gamma\mathcal{T}_{\mathbf{G}}(\gamma)\right)^t \circ \mathbf{G} \right\rangle$$

$$\preceq \mathbf{G} + \alpha\gamma^2 \mathbf{G} \cdot \left\langle \mathbf{G}, \frac{1}{\gamma}\left(\mathbf{I} - (\mathbf{I} - \gamma\mathbf{G})^{2n}\right) \right\rangle \quad \text{(by Lemma D.1)}$$

$$\preceq \mathbf{G} + \alpha\gamma^2 \left\langle \mathbf{G}, \frac{1}{\gamma}\mathbf{I} \right\rangle = \left(1 + \alpha\gamma \operatorname{tr}(\mathbf{G})\right) \cdot \mathbf{G}.$$

We now prove the second conclusion by induction. For $n = 1$, it holds because of Lemma D.1:

$$\left(\mathcal{I} - \gamma\mathcal{T}_{\mathbf{G}}(\gamma)\right) \circ \mathbf{G} \preceq \sum_{t=0}^{1} \left(\mathcal{I} - \gamma\mathcal{T}_{\mathbf{G}}(\gamma)\right)^t \circ \mathbf{G} \preceq \frac{1}{\gamma} \cdot \mathbf{I}.$$

Now consider $n \geq 2$ based on (10). We bound the second term in (10) separately for $\sum_{t=0}^{n/2-1}$ and $\sum_{t=n/2}^{n-1}$. For the first part,

$$\sum_{t=0}^{n/2-1} (\mathbf{I} - \gamma\mathbf{G})^{2(n-1-t)}\mathbf{G} \cdot \left\langle \mathbf{G}, \left(\mathcal{I} - \gamma\mathcal{T}_{\mathbf{G}}(\gamma)\right)^t \circ \mathbf{G} \right\rangle$$

$$\preceq (\mathbf{I} - \gamma\mathbf{G})^n \mathbf{G} \cdot \left\langle \mathbf{G}, \sum_{t=0}^{n/2-1} \left(\mathcal{I} - \gamma\mathcal{T}_{\mathbf{G}}(\gamma)\right)^t \circ \mathbf{G} \right\rangle$$

$$\preceq (\mathbf{I} - \gamma\mathbf{G})^n \mathbf{G} \cdot \left\langle \mathbf{G}, \frac{1}{\gamma}\mathbf{I} \right\rangle \quad \text{(by Lemma D.1)}$$

$$\preceq \frac{\operatorname{tr}(\mathbf{G})}{\gamma} \cdot (\mathbf{I} - \gamma\mathbf{G})^n \mathbf{G} \preceq \frac{\operatorname{tr}(\mathbf{G})}{\gamma} \cdot \frac{1}{n\gamma} \cdot \mathbf{I}. \tag{11}$$

For the second part,

$$\sum_{t=n/2}^{n-1} (\mathbf{I} - \gamma\mathbf{G})^{2(n-1-t)}\mathbf{G} \cdot \left\langle \mathbf{G}, \left(\mathcal{I} - \gamma\mathcal{T}_{\mathbf{G}}(\gamma)\right)^t \circ \mathbf{G} \right\rangle$$

$$\preceq \sum_{t=n/2}^{n-1} (\mathbf{I} - \gamma\mathbf{G})^{2(n-1-t)}\mathbf{G} \cdot \left\langle \mathbf{G}, \frac{1}{1 - 2\alpha\gamma \operatorname{tr}(\mathbf{G})} \cdot \frac{2}{n\gamma} \cdot \mathbf{I} \right\rangle \quad \text{(by induction hypothesis)}$$

$$\preceq \frac{\operatorname{tr}(\mathbf{G})}{1 - 2\alpha\gamma \operatorname{tr}(\mathbf{G})} \cdot \frac{2}{n\gamma} \cdot \sum_{t=n/2}^{n-1} (\mathbf{I} - \gamma\mathbf{G})^{(n-1-t)}\mathbf{G}$$

$$= \frac{\operatorname{tr}(\mathbf{G})}{1 - 2\alpha\gamma \operatorname{tr}(\mathbf{G})} \cdot \frac{2}{n\gamma} \cdot \frac{1}{\gamma} \cdot \left(\mathbf{I} - (\mathbf{I} - \gamma\mathbf{G})^{n/2}\right)$$

$$\preceq \frac{\operatorname{tr}(\mathbf{G})}{1 - 2\alpha\gamma \operatorname{tr}(\mathbf{G})} \cdot \frac{2}{n\gamma} \cdot \frac{1}{\gamma} \cdot \mathbf{I}. \tag{12}$$

Inserting (11) and (12) into (10), and apply that $(\mathbf{I} - \gamma\mathbf{G})^{2n}\mathbf{G} \preceq \frac{1}{2n\gamma} \cdot \mathbf{I}$, we obtain that

$$\left(\mathcal{I} - \gamma\mathcal{T}_{\mathbf{G}}(\gamma)\right)^n \circ \mathbf{G} \preceq \frac{1}{2n\gamma} \cdot \mathbf{I} + \frac{\alpha \operatorname{tr}(\mathbf{G})}{n} \cdot \mathbf{I} + \frac{\alpha \operatorname{tr}(\mathbf{G})}{1 - 2\alpha\gamma \operatorname{tr}(\mathbf{G})} \cdot \frac{2}{n} \cdot \mathbf{I}$$

$$= \left(\frac{1}{2} + \alpha\gamma \operatorname{tr}(\mathbf{G}) + \frac{2\alpha\gamma \operatorname{tr}(\mathbf{G})}{1 - 2\alpha\gamma \operatorname{tr}(\mathbf{G})}\right) \cdot \frac{1}{n\gamma} \cdot \mathbf{I}$$

$$\preceq \frac{1}{1 - 2\alpha\gamma\operatorname{tr}(\mathbf{G})} \cdot \frac{1}{n\gamma} \cdot \mathbf{I}.$$

We have completed the induction. $\qquad\square$

**Lemma D.3** (Bounds on the summation of bias iterates). *Suppose that Assumption 1A holds. Suppose that $\gamma_0 < 1/(2\alpha\operatorname{tr}(\mathbf{G}))$. Then the following holds for every index set $\mathbb{J} \subset \bar{\mathbb{N}}_+$:*

$$\sum_{t=1}^{M_{\mathrm{eff}}} \langle \mathbf{G}, \mathbf{B}_{t-1} \rangle \leq \frac{1}{\gamma_0} \cdot \langle \mathbf{I}_{\mathbb{J}} + 2M_{\mathrm{eff}}\gamma_0 \mathbf{G}_{\mathbb{J}^c}, \ \mathbf{B}_0 \rangle.$$

*Proof.* Notice that

$$\sum_{t=1}^{M_{\mathrm{eff}}} \langle \mathbf{G}, \mathbf{B}_{t-1} \rangle = \sum_{t=1}^{M_{\mathrm{eff}}} \langle \mathbf{G}, \ (\mathcal{I} - \gamma_0\mathcal{T}_{\mathbf{G}}(\gamma_0))^{t-1} \circ \mathbf{B}_0 \rangle = \langle \sum_{t=1}^{M_{\mathrm{eff}}} (\mathcal{I} - \gamma_0\mathcal{T}_{\mathbf{G}}(\gamma_0))^{t-1} \circ \mathbf{G}, \ \mathbf{B}_0 \rangle.$$

Then we apply Lemma D.1 to obtain that

$$\sum_{t=1}^{M_{\mathrm{eff}}} (\mathcal{I} - \gamma_0 \cdot \mathcal{T}_{\mathbf{G}}(\gamma_0))^{t-1} \circ \mathbf{G} \preceq \frac{1}{\gamma_0} \cdot \left(\mathbf{I} - (\mathbf{I} - \gamma_0\mathbf{G})^{2M_{\mathrm{eff}}}\right) \preceq \frac{1}{\gamma_0} \cdot \left(\mathbf{I}_{\mathbb{J}} + 2M_{\mathrm{eff}}\gamma_0 \mathbf{G}_{\mathbb{J}^c}\right).$$

This completes the proof. $\qquad\square$

**Lemma D.4** (Crude bounds on the bias iterates). *Suppose that Assumption 1A holds. Suppose that $\gamma_0 < 1/(2\alpha\operatorname{tr}(\mathbf{G}))$. Then the following holds for every index set $\mathbb{J} \subset \bar{\mathbb{N}}_+$ and $t \geq M_{\mathrm{eff}}$:*

$$\langle \mathbf{G}, \mathbf{B}_t \rangle \leq \frac{e}{1 - 2\alpha\operatorname{tr}(\mathbf{G})\gamma_0} \cdot \left\langle \frac{1}{M_{\mathrm{eff}}\gamma_0} \cdot \mathbf{I}_{0:j} + \mathbf{G}_{j:\infty}, \ \mathbf{B}_0 \right\rangle.$$

*Proof.* Let $L(t) = \lfloor t\log(M)/M \rfloor = \lfloor t/M_{\mathrm{eff}} \rfloor$, then $L(t) \geq 1$ as $t \geq M_{\mathrm{eff}}$. Notice that

$$\langle \mathbf{G}, \mathbf{B}_t \rangle := \left\langle \mathbf{G}, \ \prod_{t=0}^{t-1} (\mathcal{I} - \gamma_t \cdot \mathcal{T}_{\mathbf{G}}(\gamma_t)) \circ \mathbf{B}_0 \right\rangle$$

$$= \left\langle \mathbf{G}, \ \left(\mathcal{I} - \frac{\gamma_0}{2^{L(t)}} \cdot \mathcal{T}_{\mathbf{G}}\left(\frac{\gamma_0}{2^{L(t)}}\right)\right)^{t-L(t)\log(M)} \circ \prod_{\ell=0}^{L(t)-1} \left(\mathcal{I} - \frac{\gamma_0}{2^\ell} \cdot \mathcal{T}_{\mathbf{G}}\left(\frac{\gamma_0}{2^\ell}\right)\right)^{M_{\mathrm{eff}}} \circ \mathbf{B}_0 \right\rangle$$

$$= \left\langle \prod_{\ell=L(t)-1}^{0} \left(\mathcal{I} - \frac{\gamma_0}{2^\ell} \cdot \mathcal{T}_{\mathbf{G}}\left(\frac{\gamma_0}{2^\ell}\right)\right)^{M_{\mathrm{eff}}} \circ \left(\mathcal{I} - \frac{\gamma_0}{2^{L(t)}} \cdot \mathcal{T}_{\mathbf{G}}\left(\frac{\gamma_0}{2^{L(t)}}\right)\right)^{t-L(t)\log(M)} \circ \mathbf{G}, \ \mathbf{B}_0 \right\rangle.$$

We then recursively use Lemma D.2 to obtain that

$$\prod_{\ell=L(t)-1}^{0} \left(\mathcal{I} - \frac{\gamma_0}{2^\ell} \cdot \mathcal{T}_{\mathbf{G}}\left(\frac{\gamma_0}{2^\ell}\right)\right)^{M_{\mathrm{eff}}} \circ \left(\mathcal{I} - \frac{\gamma_0}{2^{L(t)}} \cdot \mathcal{T}_{\mathbf{G}}\left(\frac{\gamma_0}{2^{L(t)}}\right)\right)^{t-L(t)\log(M)} \circ \mathbf{G}$$

$$\preceq \left(1 + \frac{\gamma_0}{2^{L(t)}} \cdot \alpha\operatorname{tr}(\mathbf{G})\right) \cdot \prod_{\ell=L(t)-1}^{0} \left(\mathcal{I} - \frac{\gamma_0}{2^\ell} \cdot \mathcal{T}_{\mathbf{G}}\left(\frac{\gamma_0}{2^\ell}\right)\right)^{M_{\mathrm{eff}}} \circ \mathbf{G}$$

$$\preceq \prod_{\ell=1}^{L(t)} \left(1 + \frac{\gamma_0}{2^\ell} \cdot \alpha\operatorname{tr}(\mathbf{G})\right) \cdot (\mathcal{I} - \gamma_0 \cdot \mathcal{T}_{\mathbf{G}}(\gamma_0))^{M_{\mathrm{eff}}} \circ \mathbf{G}$$

$$\preceq e^{\alpha\operatorname{tr}(\mathbf{G})\sum_{\ell=1}^{L(t)} \gamma_0/2^\ell} \cdot (\mathcal{I} - \gamma_0 \cdot \mathcal{T}_{\mathbf{G}}(\gamma_0))^{M_{\mathrm{eff}}} \circ \mathbf{G}$$

$$\preceq e^{\alpha\operatorname{tr}(\mathbf{G})\gamma_0} \cdot (\mathcal{I} - \gamma_0 \cdot \mathcal{T}_{\mathbf{G}}(\gamma_0))^{M_{\mathrm{eff}}} \circ \mathbf{G}$$

$$\preceq e \cdot (\mathcal{I} - \gamma_0 \cdot \mathcal{T}_{\mathbf{G}}(\gamma_0))^{M_{\mathrm{eff}}} \circ \mathbf{G} \preceq \frac{e}{1 - 2\alpha\operatorname{tr}(\mathbf{G})\gamma_0} \cdot \left(\frac{1}{M_{\mathrm{eff}}\gamma_0} \cdot \mathbf{I}_{\mathbb{J}} + \mathbf{G}_{\mathbb{J}^c}\right).$$

Combining everything we complete the proof. $\qquad\square$

**Lemma D.5** (Upper bounds for the bias iterates). *Suppose that Assumption 1A holds. Suppose that* $\gamma_0 < 1/(2\alpha \operatorname{tr}(\mathbf{G}))$. *Let* $M_{\mathtt{eff}} := M/\log(M)$. *Then the following holds for every index set* $\mathbb{J} \subset \mathbb{N}_+$:

$$\mathbf{B}_M \preceq \prod_{t=1}^{M}(\mathbf{I} - \gamma_t \mathbf{G}) \cdot \mathbf{B}_0 \cdot \prod_{t=1}^{M}(\mathbf{I} - \gamma_t \mathbf{G})$$

$$+ \frac{12e\alpha}{1 - 2\alpha \operatorname{tr}(\mathbf{G})\gamma_0} \cdot \left\langle \frac{\mathbf{I}_{\mathbb{J}}}{M_{\mathtt{eff}}\gamma_0} + \mathbf{G}_{\mathbb{J}^c}, \ \mathbf{B}_0 \right\rangle \cdot \frac{\mathbf{G}_{\mathbb{J}}^{-1} + M_{\mathtt{eff}}^2 \gamma_0^2 \cdot \mathbf{G}_{\mathbb{J}^c}}{M_{\mathtt{eff}}}.$$

*Proof.* We begin with the following inequality:

$$\mathbf{B}_{t+1} = \left(\mathcal{I} - \gamma_t \widetilde{\mathcal{T}}_{\mathbf{G}}(\gamma_t)\right) \circ \mathbf{B}_t + \gamma_t^2 \cdot (\mathcal{M}_{\mathbf{G}} - \widetilde{\mathcal{M}}_{\mathbf{G}}) \circ \mathbf{B}_t$$

$$\preceq \left(\mathcal{I} - \gamma_t \widetilde{\mathcal{T}}_{\mathbf{G}}(\gamma_t)\right) \circ \mathbf{B}_t + \alpha \gamma_t^2 \cdot \mathbf{G} \cdot \langle \mathbf{G}, \ \mathbf{B}_t \rangle,$$

which implies that

$$\mathbf{B}_M \preceq \prod_{t=0}^{M-1} \left(\mathcal{I} - \gamma_t \widetilde{\mathcal{T}}_{\mathbf{G}}(\gamma_t)\right) \circ \mathbf{B}_0 + \alpha \cdot \sum_{t=0}^{M-1} \gamma_t^2 \cdot \prod_{i=t+1}^{M-1} \left(\mathcal{I} - \gamma_i \widetilde{\mathcal{T}}_{\mathbf{G}}(\gamma_i)\right) \circ \mathbf{G} \cdot \langle \mathbf{G}, \ \mathbf{B}_t \rangle$$

$$= \prod_{t=0}^{M-1} \left(\mathcal{I} - \gamma_t \widetilde{\mathcal{T}}_{\mathbf{G}}(\gamma_t)\right) \circ \mathbf{B}_0 + \alpha \cdot \sum_{t=0}^{M-1} \gamma_t^2 \cdot \prod_{i=t+1}^{M} (\mathbf{I} - \gamma_i \mathbf{G})^2 \mathbf{G} \cdot \langle \mathbf{G}, \ \mathbf{B}_t \rangle. \tag{13}$$

We next bound the second term in (13) separately for $\sum_{t=0}^{M_{\mathtt{eff}}-1}(\cdot)$ and $\sum_{t=M_{\mathtt{eff}}}^{M-1}(\cdot)$. For the first part,

$$\sum_{t=0}^{M_{\mathtt{eff}}-1} \gamma_t^2 \cdot \prod_{i=t+1}^{M} (\mathbf{I} - \gamma_i \mathbf{G})^2 \mathbf{G} \cdot \langle \mathbf{G}, \ \mathbf{B}_t \rangle = \gamma_0^2 \cdot \sum_{t=0}^{M_{\mathtt{eff}}-1} \prod_{i=t+1}^{M} (\mathbf{I} - \gamma_i \mathbf{G})^2 \mathbf{G} \cdot \langle \mathbf{G}, \ \mathbf{B}_t \rangle$$

$$\leq \gamma_0^2 \cdot \sum_{t=0}^{M_{\mathtt{eff}}-1} \prod_{i=M_{\mathtt{eff}}}^{2M_{\mathtt{eff}}-1} (\mathbf{I} - \gamma_i \mathbf{G})^2 \mathbf{G} \cdot \langle \mathbf{G}, \ \mathbf{B}_t \rangle = \gamma_0^2 \cdot \left(\mathbf{I} - \frac{\gamma_0}{2}\mathbf{G}\right)^{2M_{\mathtt{eff}}} \mathbf{G} \cdot \sum_{t=0}^{M_{\mathtt{eff}}-1} \langle \mathbf{G}, \ \mathbf{B}_t \rangle$$

$$\leq \gamma_0 \cdot \left(\mathbf{I} - \frac{\gamma_0}{2}\mathbf{G}\right)^{2M_{\mathtt{eff}}} \mathbf{G} \cdot \langle \mathbf{I}_{\mathbb{J}} + 2M_{\mathtt{eff}}\gamma_0 \mathbf{G}_{\mathbb{J}^c}, \ \mathbf{B}_0 \rangle \qquad \text{(by Lemma D.3)}$$

$$\leq \frac{8}{M_{\mathtt{eff}}^2 \gamma_0} \cdot \left(\mathbf{G}_{\mathbb{J}}^{-1} + M_{\mathtt{eff}}^2 \gamma_0^2 \mathbf{G}_{\mathbb{J}^c}\right) \cdot \langle \mathbf{I}_{\mathbb{J}} + M_{\mathtt{eff}}\gamma_0 \mathbf{G}_{\mathbb{J}^c}, \ \mathbf{B}_0 \rangle, \tag{14}$$

where in the last inequality we use that

$$\left(\mathbf{I} - \frac{\gamma_0}{2}\mathbf{G}\right)^{2M_{\mathtt{eff}}} \preceq \left(\frac{2}{M_{\mathtt{eff}}\gamma_0}\mathbf{G}_{\mathbb{J}}^{-1} + \mathbf{I}_{\mathbb{J}^c}\right)^2 \preceq 4 \cdot \left(\frac{1}{M_{\mathtt{eff}}^2 \gamma_0^2}\mathbf{G}_{\mathbb{J}}^{-2} + \mathbf{I}_{\mathbb{J}^c}\right).$$

For the second part, we apply Lemma D.4 to obtain that

$$\sum_{t=M_{\mathtt{eff}}}^{M-1} \gamma_t^2 \cdot \prod_{i=t+1}^{M} (\mathbf{I} - \gamma_i \mathbf{G})^2 \mathbf{G} \cdot \langle \mathbf{G}, \ \mathbf{B}_t \rangle$$

$$\leq \frac{e}{1 - 2\alpha \operatorname{tr}(\mathbf{G})\gamma_0} \cdot \left\langle \frac{1}{M_{\mathtt{eff}}\gamma_0} \cdot \mathbf{I}_{\mathbb{J}} + \mathbf{G}_{\mathbb{J}^c}, \ \mathbf{B}_0 \right\rangle \cdot \sum_{t=M_{\mathtt{eff}}}^{M-1} \gamma_t^2 \cdot \prod_{i=t+1}^{M} (\mathbf{I} - \gamma_i \mathbf{G})^2 \mathbf{G}$$

$$\leq \frac{8e}{1 - 2\alpha \operatorname{tr}(\mathbf{G})\gamma_0} \cdot \left\langle \frac{1}{M_{\mathtt{eff}}\gamma_0} \cdot \mathbf{I}_{\mathbb{J}} + \mathbf{G}_{\mathbb{J}^c}, \ \mathbf{B}_0 \right\rangle \cdot \left(\frac{1}{M_{\mathtt{eff}}}\mathbf{G}_{\mathbb{J}}^{-1} + M_{\mathtt{eff}}\gamma_0^2 \cdot \mathbf{G}_{\mathbb{J}^c}\right), \tag{15}$$

where in the last inequality we use Lemma C.2 (by setting $\mathbf{H}$ to $\mathbf{G}$).

Finally, inserting (14) and (15) into (13) completes the proof. $\qquad \square$

**Theorem D.1** (Bias error upper bound). *Suppose that Assumption 1A holds. Suppose that* $\gamma_0 < \min\{1/(4\alpha \operatorname{tr}(\mathbf{G})), \ 1/(\alpha \operatorname{tr}(\mathbf{H}))\}$, $\gamma_M < 1/(4\alpha \operatorname{tr}(\mathbf{H}))$. *Let* $M_{\mathtt{eff}} := M/\log(M)$, $N_{\mathtt{eff}} := N/\log(N)$. *Then it holds that*

$$\langle \mathbf{H}, \mathbf{B}_{M+N} \rangle \leq \left\| \prod_{t=M}^{M+N-1} (\mathbf{I} - \gamma_t \mathbf{H}) \prod_{t=0}^{M-1} (\mathbf{I} - \gamma_t \mathbf{G})(\mathbf{w}_0 - \mathbf{w}^*) \right\|_{\mathbf{H}}^2$$

$$+ 24e\alpha \cdot \left\| \mathbf{w}_0 - \mathbf{w}^* \right\|^2_{\frac{\mathbf{I}_{\mathbb{J}}}{M_{\texttt{eff}}\gamma_0} + \mathbf{G}_{\mathbb{J}^c}} \cdot \frac{D_{\texttt{eff}}^{\texttt{finetune}}}{M_{\texttt{eff}}}$$

$$+ 576e^2\alpha \cdot \left( \left\| \prod_{t=0}^{M-1} (\mathbf{I} - \gamma_t\mathbf{G})(\mathbf{w}_0 - \mathbf{w}^*) \right\|^2_{\frac{\mathbf{I}_{\mathbb{K}}}{N_{\texttt{eff}}\gamma_M} + \mathbf{H}_{\mathbb{K}^c}} + \left\| \mathbf{w}_0 - \mathbf{w}^* \right\|^2_{\frac{\mathbf{I}_{\mathbb{J}}}{M_{\texttt{eff}}\gamma_0} + \mathbf{G}_{\mathbb{J}^c}} \right) \cdot \frac{D_{\texttt{eff}}}{N_{\texttt{eff}}},$$

*where*

$$D_{\texttt{eff}} := \text{tr}(\mathbf{H}\mathbf{H}_{\mathbb{K}}^{-1}) + N_{\texttt{eff}}^2\gamma_M^2 \cdot \text{tr}(\mathbf{H}\mathbf{H}_{\mathbb{K}^c}),$$

$$D_{\texttt{eff}}^{\texttt{finetune}} := \text{tr}\left( \prod_{t=0}^{N-1} (\mathbf{I} - \gamma_{M+t}\mathbf{H})^2\mathbf{H} \cdot \left( \mathbf{G}_{\mathbb{J}}^{-1} + M_{\texttt{eff}}^2\gamma_0^2 \cdot \mathbf{G}_{\mathbb{J}^c} \right) \right),$$

*and $\mathbb{K}$, $\mathbb{J}$ can be arbitrary index sets.*

*Proof.* We first apply Lemma D.5 by setting $\mathbf{B}_0$ to $\mathbf{B}_M$, $\gamma_0$ to $\gamma_M$, $M$ to $N$ and $\mathbf{G}$ to $\mathbf{H}$, so that we have

$$\mathbf{B}_{M+N} \preceq \prod_{t=0}^{N-1} (\mathbf{I} - \gamma_{M+t}\mathbf{H})\mathbf{B}_M \prod_{t=0}^{N-1} (\mathbf{I} - \gamma_{M+t}\mathbf{H})$$

$$+ 24e\alpha \cdot \left\langle \frac{\mathbf{I}_{\mathbb{K}}}{N_{\texttt{eff}}\gamma_M} + \mathbf{H}_{\mathbb{K}^c}, \mathbf{B}_M \right\rangle \cdot \frac{\mathbf{H}_{\mathbb{K}}^{-1} + N_{\texttt{eff}}^2\gamma_M^2 \cdot \mathbf{H}_{\mathbb{K}^c}}{N_{\texttt{eff}}}.$$

Taking inner product with $\mathbf{H}$ we obtain

$$\langle \mathbf{H}, \mathbf{B}_{M+N} \rangle \preceq \left\langle \mathbf{H}, \prod_{t=0}^{N-1} (\mathbf{I} - \gamma_{M+t}\mathbf{H})\mathbf{B}_M \prod_{t=0}^{N-1} (\mathbf{I} - \gamma_{M+t}\mathbf{H}) \right\rangle$$

$$+ 24e\alpha \cdot \left\langle \frac{\mathbf{I}_{\mathbb{K}}}{N_{\texttt{eff}}\gamma_M} + \mathbf{H}_{\mathbb{K}^c}, \mathbf{B}_M \right\rangle \cdot \frac{D_{\texttt{eff}}}{N_{\texttt{eff}}}$$

$$= \left\langle \prod_{t=0}^{N-1} (\mathbf{I} - \gamma_{M+t}\mathbf{H})^2\mathbf{H}, \mathbf{B}_M \right\rangle + +24e\alpha \cdot \left\langle \frac{\mathbf{I}_{\mathbb{K}}}{N_{\texttt{eff}}\gamma_M} + \mathbf{H}_{\mathbb{K}^c}, \mathbf{B}_M \right\rangle \cdot \frac{D_{\texttt{eff}}}{N_{\texttt{eff}}}.$$

Now applying the upper bound for $\mathbf{B}_M$ in Lemma D.5, we obtain

$$\langle \mathbf{H}, \mathbf{B}_{M+N} \rangle$$

$$\preceq \underbrace{\left\langle \prod_{t=0}^{N-1} (\mathbf{I} - \gamma_{M+t}\mathbf{H})^2\mathbf{H}, \prod_{t=0}^{M-1} (\mathbf{I} - \gamma_t\mathbf{G})\mathbf{B}_0 \prod_{t=1}^{M} (\mathbf{I} - \gamma_t\mathbf{G}) \right\rangle}_{(\spadesuit)}$$

$$+ 24e\alpha \cdot \underbrace{\left\langle \frac{\mathbf{I}_{\mathbb{J}}}{M_{\texttt{eff}}\gamma_0} + \mathbf{G}_{\mathbb{J}^c}, \mathbf{B}_0 \right\rangle}_{(\heartsuit)} \cdot \frac{D_{\texttt{eff}}^{\texttt{finetune}}}{M_{\texttt{eff}}}$$

$$+ 24e\alpha \cdot \underbrace{\left\langle \frac{\mathbf{I}_{\mathbb{K}}}{N_{\texttt{eff}}\gamma_M} + \mathbf{H}_{\mathbb{K}^c}, \prod_{t=0}^{M-1} (\mathbf{I} - \gamma_t\mathbf{G})\mathbf{B}_0 \prod_{t=1}^{M} (\mathbf{I} - \gamma_t\mathbf{G}) \right\rangle}_{(\diamondsuit)} \cdot \frac{D_{\texttt{eff}}}{N_{\texttt{eff}}}$$

$$+ 576e^2\alpha \cdot \underbrace{\left\langle \frac{\mathbf{I}_{\mathbb{J}}}{M_{\texttt{eff}}\gamma_0} + \mathbf{G}_{\mathbb{J}^c}, \mathbf{B}_0 \right\rangle}_{(\heartsuit)} \cdot \alpha \underbrace{\left\langle \frac{\mathbf{I}_{\mathbb{K}}}{N_{\texttt{eff}}\gamma_M} + \mathbf{H}_{\mathbb{K}^c}, \frac{\mathbf{G}_{\mathbb{J}}^{-1} + M_{\texttt{eff}}^2\gamma_0^2 \cdot \mathbf{G}_{\mathbb{J}^c}}{M_{\texttt{eff}}} \right\rangle}_{(\clubsuit)} \cdot \frac{D_{\texttt{eff}}}{N_{\texttt{eff}}}.$$

By definition we know that

$$(\spadesuit) = \left\| \prod_{t=M}^{M+N-1} (\mathbf{I} - \gamma_t\mathbf{H}) \prod_{t=0}^{M-1} (\mathbf{I} - \gamma_t\mathbf{G})(\mathbf{w}_0 - \mathbf{w}^*) \right\|^2_{\mathbf{H}},$$

$$(\heartsuit) = \left\| \mathbf{w}_0 - \mathbf{w}^* \right\|^2_{\frac{\mathbf{I}_{\mathbb{J}}}{M_{\texttt{eff}}\gamma_0} + \mathbf{G}_{\mathbb{J}^c}},$$

$$(\lozenge) = \Big\| \prod_{t=0}^{M-1} (\mathbf{I} - \gamma_t \mathbf{G})(\mathbf{w}_0 - \mathbf{w}^*) \Big\|^2_{\frac{\mathbf{I}_{\mathbb{K}}}{N_{\texttt{eff}} \gamma_M} + \mathbf{H}_{\mathbb{K}^c}}.$$

As for ($\clubsuit$), we can choose $\mathbb{K} = \emptyset$ and $\mathbb{J} = \{j : \mu_j \geq 1/(M_{\texttt{eff}} \gamma_0)\}$ so that

$$(\clubsuit) \leq \alpha \langle \mathbf{H}, \ \gamma_0 \mathbf{I} \rangle \leq \alpha \gamma_0 \operatorname{tr}(\mathbf{H}) \leq 1.$$

Putting everything together completes the proof. $\qquad \square$

## D.2   Lower Bounds

**Lemma D.6** (Lower bounds for the bias iterates). *Suppose that Assumption 1B holds. Suppose that $\gamma_0 < 1/\|\mathbf{G}\|_2$. Let $M_{\texttt{eff}} := M/\log(M)$ and suppose that $M_{\texttt{eff}} \geq 10$. Let $\mathbb{J} := \{j : \mu_j \geq 1/(M_{\texttt{eff}} \gamma_0)\}$. Then it holds that*

$$\mathbf{B}_M \succeq \prod_{t=0}^{M-1} (\mathbf{I} - \gamma_t \mathbf{G}) \mathbf{B}_0 \prod_{t=1}^{M} (\mathbf{I} - \gamma_t \mathbf{G}) + \frac{\beta}{1200} \cdot \langle \mathbf{G}_{\mathbb{J}^c}, \ \mathbf{B}_0 \rangle \cdot \frac{\mathbf{G}_{\mathbb{J}}^{-1} + M_{\texttt{eff}}^2 \gamma_0^2 \cdot \mathbf{G}_{\mathbb{J}^c}}{M_{\texttt{eff}}}.$$

*Proof.* This is from Theorem 8 in Wu et al. [2021]. $\qquad \square$

**Theorem D.2** (Lower bounds for the bias error). *Suppose that Assumption 1B holds. Suppose that $\gamma_0 < 1/\|\mathbf{G}\|_2$, $\gamma_M < 1/\|\mathbf{H}\|_2$. Let $M_{\texttt{eff}} := M/\log(M)$, $N_{\texttt{eff}} := N/\log(N)$, and suppose that $M_{\texttt{eff}}, N_{\texttt{eff}} \geq 10$. Let $\mathbb{J} := \{j : \mu_j \geq 1/(M_{\texttt{eff}} \gamma_0)\}$, $\mathbb{K} := \{k : \lambda_k \geq 1/(N_{\texttt{eff}} \gamma_M)\}$. Then it holds that*

$$\langle \mathbf{H}, \mathbf{B}_{M+N} \rangle \geq \Big\| \prod_{t=M}^{M+N-1} (\mathbf{I} - \gamma_t \mathbf{H}) \prod_{t=0}^{M-1} (\mathbf{I} - \gamma_t \mathbf{G})(\mathbf{w}_0 - \mathbf{w}^*) \Big\|^2_{\mathbf{H}}$$

$$+ \frac{\beta}{1200} \cdot \|\mathbf{w}_0 - \mathbf{w}^*\|^2_{\mathbf{G}_{\mathbb{J}^c}} \cdot \frac{D_{\texttt{eff}}^{\texttt{finetune}}}{M_{\texttt{eff}}} + \frac{\beta}{1200} \cdot \Big\| \prod_{t=0}^{M-1} (\mathbf{I} - \gamma_t \mathbf{G})(\mathbf{w}_0 - \mathbf{w}^*) \Big\|^2_{\mathbf{H}_{\mathbb{K}^c}} \cdot \frac{D_{\texttt{eff}}}{N_{\texttt{eff}}},$$

*where*

$$D_{\texttt{eff}} := \operatorname{tr}(\mathbf{H} \mathbf{H}_{\mathbb{K}}^{-1}) + N_{\texttt{eff}}^2 \gamma_M^2 \cdot \operatorname{tr}(\mathbf{H} \mathbf{H}_{\mathbb{K}^c}),$$

$$D_{\texttt{eff}}^{\texttt{finetune}} := \operatorname{tr} \Big( \prod_{t=0}^{N-1} (\mathbf{I} - \gamma_{M+t} \mathbf{H})^2 \mathbf{H} \cdot \big( \mathbf{G}_{\mathbb{J}}^{-1} + M_{\texttt{eff}}^2 \gamma_0^2 \cdot \mathbf{G}_{\mathbb{J}^c} \big) \Big).$$

*Proof.* We first apply Lemma D.6 by setting $\mathbf{B}_0$ to $\mathbf{B}_M$, $\gamma_0$ to $\gamma_M$, $M$ to $N$ and $\mathbf{G}$ to $\mathbf{H}$, so that we have

$$\mathbf{B}_{M+N} \succeq \prod_{t=0}^{N-1} (\mathbf{I} - \gamma_{M+t} \mathbf{H}) \mathbf{B}_M \prod_{t=0}^{N-1} (\mathbf{I} - \gamma_{M+t} \mathbf{H}) + \frac{\beta}{1200} \langle \mathbf{H}_{\mathbb{K}^c}, \mathbf{B}_M \rangle \frac{\mathbf{H}_{\mathbb{K}}^{-1} + N_{\texttt{eff}}^2 \gamma_M^2 \cdot \mathbf{H}_{\mathbb{K}^c}}{N_{\texttt{eff}}}.$$

Taking inner product with $\mathbf{H}$ we obtain

$$\langle \mathbf{H}, \mathbf{B}_{M+N} \rangle \succeq \Big\langle \mathbf{H}, \ \prod_{t=0}^{N-1} (\mathbf{I} - \gamma_{M+t} \mathbf{H}) \mathbf{B}_M \prod_{t=0}^{N-1} (\mathbf{I} - \gamma_{M+t} \mathbf{H}) \Big\rangle + \frac{\beta}{1200} \cdot \langle \mathbf{H}_{\mathbb{K}^c}, \mathbf{B}_M \rangle \cdot \frac{D_{\texttt{eff}}}{N_{\texttt{eff}}}$$

$$= \Big\langle \prod_{t=0}^{N-1} (\mathbf{I} - \gamma_{M+t} \mathbf{H})^2 \mathbf{H}, \ \mathbf{B}_M \Big\rangle + \frac{\beta}{1200} \cdot \langle \mathbf{H}_{\mathbb{K}^c}, \mathbf{B}_M \rangle \cdot \frac{D_{\texttt{eff}}}{N_{\texttt{eff}}}.$$

Now applying the lower bound for $\mathbf{B}_M$ in Lemma D.6, we obtain

$$\langle \mathbf{H}, \mathbf{B}_{M+N} \rangle \succeq \Big\langle \prod_{t=0}^{N-1} (\mathbf{I} - \gamma_{M+t} \mathbf{H})^2 \mathbf{H}, \ \prod_{t=0}^{M-1} (\mathbf{I} - \gamma_t \mathbf{G}) \mathbf{B}_0 \prod_{t=1}^{M} (\mathbf{I} - \gamma_t \mathbf{G}) \Big\rangle$$

$$+ \frac{\beta}{1200} \cdot \langle \mathbf{G}_{\mathbb{J}^c}, \mathbf{B}_0 \rangle \cdot \Big\langle \prod_{t=0}^{N-1} (\mathbf{I} - \gamma_{M+t} \mathbf{H})^2 \mathbf{H}, \ \frac{\mathbf{G}_{\mathbb{J}}^{-1} + M_{\texttt{eff}}^2 \gamma_0^2 \cdot \mathbf{G}_{\mathbb{J}^c}}{M_{\texttt{eff}}} \Big\rangle$$

$$+ \frac{\beta}{1200} \cdot \left\langle \mathbf{H}_{\mathbb{K}^c}, \prod_{t=0}^{M-1} (\mathbf{I} - \gamma_t \mathbf{G}) \mathbf{B}_0 \prod_{t=1}^{M} (\mathbf{I} - \gamma_t \mathbf{G}) \right\rangle \cdot \frac{D_{\text{eff}}}{N_{\text{eff}}}$$

$$= \left\| \prod_{t=M}^{M+N-1} (\mathbf{I} - \gamma_t \mathbf{H}) \prod_{t=0}^{M-1} (\mathbf{I} - \gamma_t \mathbf{G})(\mathbf{w}_0 - \mathbf{w}^*) \right\|_{\mathbf{H}}^2$$

$$+ \frac{\beta}{1200} \cdot \|\mathbf{w}_0 - \mathbf{w}^*\|_{\mathbf{G}_{\mathbb{J}^c}}^2 \cdot \frac{D_{\text{eff}}^{\texttt{finetune}}}{M_{\text{eff}}}$$

$$+ \frac{\beta}{1200} \cdot \left\| \prod_{t=0}^{M-1} (\mathbf{I} - \gamma_t \mathbf{G})(\mathbf{w}_0 - \mathbf{w}^*) \right\|_{\mathbf{H}_{\mathbb{K}^c}}^2 \cdot \frac{D_{\text{eff}}}{N_{\text{eff}}},$$

which completes the proof. $\qquad\square$

# E  Proof of Theorems in Main Text

## E.1  Proof of Theorem 3.1

*Proof of Theorem 3.1.* This is by combining Theorems C.1 and D.1. $\qquad\square$

## E.2  Proof of Theorem 3.2

*Proof of Theorem 3.2.* This is by combining Theorems C.2 and D.2. $\qquad\square$

## E.3  Proof of Theorem 4.1

*Proof of Theorem 4.1.* During the proof, we use $\gamma^{\texttt{sl}}$ and $\gamma_0$ to denote the initial stepsizes for supervised learning and pretraining, respectively. Then Corollaries 3.4 and 3.3 sharply characterize the risk bounds for supervised learning and pretraining, respectively. In particular, let $\texttt{SNR} := \alpha \|\mathbf{w}^*\|_{\mathbf{G}}^2 / \sigma^2$, then we have

$$\texttt{ExcessRisk}(\mathbf{w}_{0+N^{\texttt{sl}}}) \gtrsim \left\| \prod_{t=0}^{N-1} (\mathbf{I} - \gamma_t^{\texttt{sl}} \mathbf{H})(\mathbf{w}_0 - \mathbf{w}^*) \right\|_{\mathbf{H}}^2 + \sigma^2 \cdot \frac{D_{\text{eff}}^{\texttt{sl}}}{N_{\text{eff}}^{\texttt{sl}}}, \qquad (16)$$

$$\texttt{ExcessRisk}(\mathbf{w}_{M+0}) \lesssim \left\| \prod_{t=0}^{M-1} (\mathbf{I} - \gamma_t \mathbf{G})(\mathbf{w}_0 - \mathbf{w}^*) \right\|_{\mathbf{H}}^2 + (1 + \texttt{SNR})\sigma^2 \cdot \frac{D_{\text{eff}}^{\texttt{pretrain}}}{M_{\text{eff}}}. \qquad (17)$$

Fix hyperparameters $(N_{\text{eff}}^{\texttt{sl}}, \gamma^{\texttt{sl}})$ for supervised learning, we now identify hyperparameters $(M_{\text{eff}}, \gamma_0)$ for pretraining so that its risk (17) is no larger than that of supervised learning (16) upto a constant factor. To this end, we claim that

$$\frac{D_{\text{eff}}^{\texttt{pretrain}}}{M_{\text{eff}}} \leq \frac{D_{\text{eff}}^{\texttt{sl}}}{N_{\text{eff}}^{\texttt{sl}}} \text{ given that } \gamma_0 \leq \frac{D_{\text{eff}}^{\texttt{sl}}}{N_{\text{eff}}^{\texttt{sl}} \operatorname{tr}(\mathbf{H})}. \qquad (18)$$

$$\left\| \prod_{t=0}^{M-1} (\mathbf{I} - \gamma_t \mathbf{G})(\mathbf{w}_0 - \mathbf{w}^*) \right\|_{\mathbf{H}}^2 \lesssim \left\| \prod_{t=0}^{N-1} (\mathbf{I} - \gamma_t^{\texttt{sl}} \mathbf{H})(\mathbf{w}_0 - \mathbf{w}^*) \right\|_{\mathbf{H}}^2 \qquad (19)$$

$$\text{given that } M_{\text{eff}} \gamma_0 \geq 4 N_{\text{eff}}^{\texttt{sl}} \gamma^{\texttt{sl}} \|\mathbf{H}_{0:k^*}\|_{\mathbf{G}}.$$

To prove (18), we consider the optimal index set $\mathbb{J}^* := \{ j : \mu_j \geq 1/(M_{\text{eff}} \gamma_0) \}$ as defined in Corollary 3.3, then by definition we have

$$\frac{D_{\text{eff}}^{\texttt{pretrain}}}{M_{\text{eff}}} \leq \frac{1}{M_{\text{eff}}} \cdot \sum_{j \in \mathbb{J}^*} \mathbf{H}_{jj} \cdot \frac{1}{\mu_j} + M_{\text{eff}} \gamma_0 \cdot \sum_{j \notin \mathbb{J}^*} \mathbf{H}_{jj} \cdot \mu_j$$

$$\leq \gamma_0 \cdot \sum_{j \in \mathbb{J}^*} \mathbf{H}_{jj} + \gamma_0 \cdot \sum_{j \notin \mathbb{J}^*} \mathbf{H}_{jj} = \gamma_0 \operatorname{tr}(\mathbf{H}),$$

which justifies (18).

To prove (19), we consider the bias error separately in its head part and its tail part, divided by the optimal index $k^* := \min\{k : \lambda_k \geq 1/(N_{\texttt{eff}}^{\texttt{sl}}\gamma^{\texttt{sl}})\}$ as defined in Corollary 3.4. For a tail index $k > k^*$, we have

$$\prod_{t=0}^{M-1}(1-\gamma_t\mu_k)^2 \leq 1 \leq 100 \cdot (1-2\gamma^{\texttt{sl}}\lambda_k)^{2N_{\texttt{eff}}^{\texttt{sl}}} \leq 100 \cdot \prod_{t=0}^{N^{\texttt{sl}}-1}(1-\gamma_t^{\texttt{sl}}\lambda_k)^2, \qquad (20)$$

where the second inequality is because that $\gamma^{\texttt{sl}}\lambda_k \geq 1/N_{\texttt{eff}}^{\texttt{sl}}$ and $N_{\texttt{eff}}^{\texttt{sl}} \geq 10$. For a head index $k \leq k^*$, we have

$$\prod_{t=0}^{M-1}(1-\gamma_t\mu_k)^2 \leq (1-\gamma_0\mu_k)^{2M_{\texttt{eff}}} \leq (1-2\gamma^{\texttt{sl}}\lambda_k)^{2N_{\texttt{eff}}^{\texttt{sl}}} \leq \prod_{t=0}^{N^{\texttt{sl}}-1}(1-\gamma_t^{\texttt{sl}}\lambda_k)^2, \qquad (21)$$

where the second inequality is because:

$$(1-\gamma_0\mu_k)^{\frac{M_{\texttt{eff}}}{N_{\texttt{eff}}^{\texttt{sl}}}} \leq 1 - \frac{M_{\texttt{eff}}}{2N_{\texttt{eff}}^{\texttt{sl}}} \cdot \gamma_0\mu_k \qquad \text{(since } ab \leq (a+b)/2 \text{ for } 0 < a,b < 1\text{)}$$

$$\leq 1 - \frac{M_{\texttt{eff}}}{2N_{\texttt{eff}}^{\texttt{sl}}\|\mathbf{H}_{0:k^*}\|_{\mathbf{G}}} \cdot \gamma_0\lambda_k \qquad \text{(since } \|\mathbf{H}_{0:k^*}\|_{\mathbf{G}} := \max\{\lambda_k/\mu_k : k \leq k^*\}\text{)}$$

$$\leq 1 - 2\gamma^{\texttt{sl}}\lambda_k. \qquad \text{(use the condition in (19))}$$

Combining (20), (21) and the definition of bias error justifies (19).

Finally, we choose $\gamma_0 = D_{\texttt{eff}}^{\texttt{sl}}/(N_{\texttt{eff}}^{\texttt{sl}}\operatorname{tr}(\mathbf{H}))$ and

$$M_{\texttt{eff}} \geq (N_{\texttt{eff}}^{\texttt{sl}})^2 \cdot \frac{4\|\mathbf{H}_{0:k^*}\|_{\mathbf{G}}}{\alpha D_{\texttt{eff}}} \geq (N_{\texttt{eff}}^{\texttt{sl}})^2 \cdot \frac{4\gamma^{\texttt{sl}}\operatorname{tr}(\mathbf{H})\|\mathbf{H}_{0:k^*}\|_{\mathbf{G}}}{D_{\texttt{eff}}^{\texttt{sl}}} = \frac{4N_{\texttt{eff}}^{\texttt{sl}}\gamma^{\texttt{sl}}\|\mathbf{H}_{0:k^*}\|_{\mathbf{G}}}{\gamma_0},$$

so that both (18) and (21) hold, which imply that the risk of pretraining (17) is no lager than of supervised learning (16) upto a constant factor.

$$\square$$

### E.4 Proof of Theorem 4.2

*Proof of Theorem 4.2.* During the proof, we use $\gamma^{\texttt{sl}}$, $\gamma_0$ and $\gamma_M$ to denote the initial stepsizes for supervised learning, pretraining and finetuning, respectively. Then Corollary 3.4 and Theorem 3.1 sharply characterize the risk bounds for supervised learning and pretraining-finetuning, respectively. In particular, let $\texttt{SNR} := \alpha(\|\mathbf{w}^*\|_{\mathbf{G}}^2 + \|\mathbf{w}^*\|_{\mathbf{H}}^2)/\sigma^2$, then we have the following upper bound for pretraining-finetuning:

$$\begin{aligned}\texttt{ExcessRisk}(\mathbf{w}_{M+N}) \lesssim \Big\| \prod_{t=M}^{M+N-1}(\mathbf{I}-\gamma_t\mathbf{H})\prod_{t=0}^{M-1}(\mathbf{I}-\gamma_t\mathbf{G})(\mathbf{w}_0-\mathbf{w}^*)\Big\|_{\mathbf{H}}^2 \\ + (1+\texttt{SNR})\sigma^2 \cdot \Big(\frac{D_{\texttt{eff}}^{\texttt{finetune}}}{M_{\texttt{eff}}} + \frac{D_{\texttt{eff}}}{N_{\texttt{eff}}}\Big),\end{aligned} \qquad (22)$$

and we have a lower bound for supervised learning shown in (16). Fix hyperparameters $(N_{\texttt{eff}}^{\texttt{sl}},\gamma^{\texttt{sl}})$ for supervised learning, we now identify hyperparameters $(M_{\texttt{eff}}, N_{\texttt{eff}}, \gamma_0, \gamma_M)$ for pretraining-finetuning so that its risk (22) is no larger than that of supervised learning (16) upto a constant factor. To this end, we claim that

$$\frac{D_{\texttt{eff}}}{N_{\texttt{eff}}} \leq \frac{D_{\texttt{eff}}^{\texttt{sl}}}{N_{\texttt{eff}}^{\texttt{sl}}} \text{ given that } \gamma_M \leq \frac{D_{\texttt{eff}}^{\texttt{sl}}}{N_{\texttt{eff}}^{\texttt{sl}}\operatorname{tr}(\mathbf{H})} \qquad (23)$$

$$\frac{D_{\texttt{eff}}^{\texttt{finetune}}}{M_{\texttt{eff}}} \leq \frac{D_{\texttt{eff}}^{\texttt{sl}}}{N_{\texttt{eff}}^{\texttt{sl}}} \text{ given that } \gamma_0 \leq \frac{D_{\texttt{eff}}^{\texttt{sl}}}{N_{\texttt{eff}}^{\texttt{sl}}\operatorname{tr}\big(\prod_{t=0}^{N-1}(\mathbf{I}-\gamma_{M+t}\mathbf{H})^2\mathbf{H}\big)}. \qquad (24)$$

$$\Big\|\prod_{t=M}^{M+N-1}(\mathbf{I}-\gamma_t\mathbf{H})\prod_{t=0}^{M-1}(\mathbf{I}-\gamma_t\mathbf{G})\mathbf{w}^*\Big\|_{\mathbf{H}}^2 \lesssim \Big\|\prod_{t=0}^{N-1}(\mathbf{I}-\gamma_t^{\texttt{sl}}\mathbf{H})\mathbf{w}^*\Big\|_{\mathbf{H}}^2 + \|\mathbf{w}^*\|_{\mathbf{H}}^2 \cdot \frac{D_{\texttt{eff}}^{\texttt{sl}}}{N_{\texttt{eff}}^{\texttt{sl}}} \qquad (25)$$

$$\text{given that } M_{\texttt{eff}}\gamma_0 \geq 4N_{\texttt{eff}}^{\texttt{sl}}\gamma^{\texttt{sl}}\|\mathbf{H}_{k^\dagger:k^*}\|_{\mathbf{G}}.$$

To prove (23) and (24), one only needs to repeat the proof for (18).

To prove (25), we consider the bias error separately in its head part, middle part and tail part, divided by the index

$$k^\dagger := \{k : \lambda_k \geq \log(N_{\mathtt{eff}}^{\mathtt{sl}})/(N_{\mathtt{eff}}\gamma_M)\}$$

and the index $k^* := \min\{k : \lambda_k \geq 1/(N_{\mathtt{eff}}^{\mathtt{sl}}\gamma^{\mathtt{sl}})\}$ as defined in Corollary 3.4. For a tail index $k > k^*$, we have

$$\prod_{t=M}^{M+N-1}(1-\gamma_t\lambda_k)^2 \prod_{t=0}^{M-1}(1-\gamma_t\mu_k)^2 \leq 1 \leq 100\cdot(1-2\gamma^{\mathtt{sl}}\lambda_k)^{2N_{\mathtt{eff}}^{\mathtt{sl}}} \leq 100\cdot\prod_{t=0}^{N^{\mathtt{sl}}-1}(1-\gamma_t^{\mathtt{sl}}\lambda_k)^2, \quad (26)$$

where the second inequality is because that $\gamma^{\mathtt{sl}}\lambda_k \geq 1/N_{\mathtt{eff}}^{\mathtt{sl}}$ and $N_{\mathtt{eff}}^{\mathtt{sl}} \geq 10$. For a middle index $k^\dagger < k \leq k^*$, we have

$$\prod_{t=M}^{M+N-1}(1-\gamma_t\lambda_k)^2 \prod_{t=0}^{M-1}(1-\gamma_t\mu_k)^2 \leq (1-\gamma_0\mu_k)^{2M_{\mathtt{eff}}} \leq (1-2\gamma^{\mathtt{sl}}\lambda_k)^{2N_{\mathtt{eff}}^{\mathtt{sl}}} \leq \prod_{t=0}^{N^{\mathtt{sl}}-1}(1-\gamma_t^{\mathtt{sl}}\lambda_k)^2,$$
$$(27)$$

where the second inequality is because:

$$(1-\gamma_0\mu_k)^{\frac{M_{\mathtt{eff}}}{N_{\mathtt{eff}}^{\mathtt{sl}}}} \leq 1 - \frac{M_{\mathtt{eff}}}{2N_{\mathtt{eff}}^{\mathtt{sl}}}\cdot\gamma_0\mu_k \qquad \text{(since } ab \leq (a+b)/2 \text{ for } 0 < a,b < 1)$$

$$\leq 1 - \frac{M_{\mathtt{eff}}\gamma_0\lambda_k}{2N_{\mathtt{eff}}^{\mathtt{sl}}\|\mathbf{H}_{k^\dagger:k^*}\|_{\mathbf{G}}} \qquad \text{(since } \|\mathbf{H}_{k^\dagger:k^*}\|_{\mathbf{G}} := \max\{\lambda_k/\mu_k : k^\dagger < k \leq k^*\})$$

$$\leq 1 - 2\gamma^{\mathtt{sl}}\lambda_k. \qquad \text{(use the condition in (25))}$$

For a head index $k \leq k^\dagger$, we have

$$\prod_{t=M}^{M+N-1}(1-\gamma_t\lambda_k)^2 \prod_{t=0}^{M-1}(1-\gamma_t\mu_k)^2 \leq (1-\gamma_M\lambda_k)^{2N_{\mathtt{eff}}} \leq e^{-2N_{\mathtt{eff}}\gamma_M\lambda_k} \leq \frac{D_{\mathtt{eff}}^{\mathtt{sl}}}{N_{\mathtt{eff}}^{\mathtt{sl}}}, \qquad (28)$$

where in the last inequality we use $\lambda_k \geq \lambda_{k^\dagger} \geq \log(N_{\mathtt{eff}}^{\mathtt{sl}})/(N_{\mathtt{eff}}\gamma_M)$.

Combining (26), (27) (28) and the definition of bias error justifies (25).

Finally, we choose

$$\gamma_0 = \gamma_M = D_{\mathtt{eff}}^{\mathtt{sl}}/(N_{\mathtt{eff}}^{\mathtt{sl}}\mathrm{tr}(\mathbf{H})),$$

$$k^\dagger := \{k : \lambda_k \geq \log(N_{\mathtt{eff}}^{\mathtt{sl}})/(N_{\mathtt{eff}}\gamma_M) = N_{\mathtt{eff}}^{\mathtt{sl}}\log(N_{\mathtt{eff}}^{\mathtt{sl}})\mathrm{tr}(\mathbf{H})/(N_{\mathtt{eff}}D_{\mathtt{eff}}^{\mathtt{sl}})\},$$

and

$$M_{\mathtt{eff}} \geq (N_{\mathtt{eff}}^{\mathtt{sl}})^2\cdot\frac{4\|\mathbf{H}_{k^\dagger:k^*}\|_{\mathbf{G}}}{\alpha D_{\mathtt{eff}}^{\mathtt{sl}}} \geq (N_{\mathtt{eff}}^{\mathtt{sl}})^2\cdot\frac{4\gamma^{\mathtt{sl}}\mathrm{tr}(\mathbf{H})\|\mathbf{H}_{k^\dagger:k^*}\|_{\mathbf{G}}}{D_{\mathtt{eff}}^{\mathtt{sl}}} = \frac{4N_{\mathtt{eff}}^{\mathtt{sl}}\gamma^{\mathtt{sl}}\|\mathbf{H}_{k^\dagger:k^*}\|_{\mathbf{G}}}{\gamma_0},$$

so that all (23), (24) and (28) hold, which imply that the risk of pretraining (22) is no lager than of supervised learning (16) upto a constant factor.

$$\square$$

## E.5 Proof of Example 4.3

*Proof of Example 4.3.* One may verify that $\mathrm{tr}(\mathbf{H}) \asymp \mathrm{tr}(\mathbf{G}) \asymp 1$ and that $\|\mathbf{w}^*\|_{\mathbf{H}}^2 \asymp \|\mathbf{w}^*\|_{\mathbf{G}}^2 \asymp \sigma^2 \asymp 1$. Therefore

$$\gamma_0 \lesssim 1/(\mathrm{tr}(\mathbf{H})) \asymp 1, \gamma_M \lesssim 1/(\mathrm{tr}(\mathbf{G})) \asymp 1.$$

**Pretraining.** From Corollary 3.3, we know that

$$\texttt{ExcessRisk}(\mathbf{w}_{M+0}) \geq \lambda_1(1-2\gamma_0\mu_1)^{2M_{\mathtt{eff}}}(\mathbf{w}^*[1])^2 \geq (1-2\gamma_0\epsilon^2)^{2M_{\mathtt{eff}}},$$

for which to be smaller than $\epsilon$ one has to set

$$M_{\mathtt{eff}} \gtrsim \gamma_0^{-1}\epsilon^{-2}\log\epsilon^{-1} \gtrsim \epsilon^{-2}.$$

**Supervised Learning.** As for supervised learning, we discuss its rate based on Corollary 3.4 and the choice of $\mathbb{K} = \{k : \lambda_k \geq 1/(N_{\text{eff}}\gamma_0)\}$.

- If $|\mathbb{K}| > \epsilon^{-0.5}$, then by Corollary 3.4 we have

$$\texttt{ExcessRisk}(\mathbf{w}_{0+N}) \gtrsim \sigma^2 \cdot \frac{|\mathbb{K}|}{N_{\text{eff}}} \gtrsim \frac{\epsilon^{-0.5}}{N_{\text{eff}}},$$

  for which to be smaller than $\epsilon$ one has to have $N_{\text{eff}} \gtrsim \epsilon^{-1.5}$.

- If $|\mathbb{K}| \leq \epsilon^{-0.5}$, then by Corollary 3.4 we have

$$\texttt{ExcessRisk}(\mathbf{w}_{0+N}) \gtrsim \sigma^2 \cdot N_{\text{eff}}\gamma_0^2 \sum_{i>k^*} \lambda_i^2 \gtrsim N_{\text{eff}}\gamma_0^2 \epsilon^{0.5},$$

  for which to be smaller than $\epsilon$ one has to have

$$N_{\text{eff}}\gamma_0^2 \lesssim \epsilon^{0.5}. \tag{29}$$

  On the other hand, by Corollary 3.4 we have

$$\texttt{ExcessRisk}(\mathbf{w}_{0+N}) \geq \lambda_2(1 - 2\gamma_0\lambda_2)^{2N_{\text{eff}}}(\mathbf{w}^*[2])^2 \geq \epsilon^{0.5} \cdot (1 - 2\gamma_0\epsilon^{0.5})^{2N_{\text{eff}}},$$

  for which to be smaller than $\epsilon$ one need to set

$$N_{\text{eff}}\gamma_0 \gtrsim \epsilon^{-0.5} \log \epsilon^{-0.5} \gtrsim \epsilon^{-0.5}. \tag{30}$$

  Then (29) and (30) together imply that $N_{\text{eff}} \gtrsim \epsilon^{-1.5}$.

In sum, for $\texttt{ExcessRisk}(\mathbf{w}_{0+N}) \leq \epsilon$ one has to set

$$N_{\text{eff}} \gtrsim \epsilon^{-1.5}.$$

**Pretraining-Finetuning.** Now we consider pretraining-finetuning by Theorem 3.1. We set

$$\gamma_0 \asymp 1, \quad \gamma_M \asymp \epsilon, \quad M_{\text{eff}} \asymp \epsilon^{-1}, \quad N_{\text{eff}} \asymp \epsilon^{-1}\log(\epsilon^{-2}). \tag{31}$$

Under (31), we see that

$$\lambda_1(1 - \gamma_M\lambda_1)^{2N_{\text{eff}}} \leq e^{-2N_{\text{eff}}\gamma_M} \lesssim \epsilon^2. \tag{32}$$

We now verify that $\texttt{ExcessRisk}(\mathbf{w}_{M+N}) \lesssim \epsilon$.

According to the proof of (18), it holds that

$$\frac{D_{\text{eff}}}{N_{\text{eff}}} \lesssim \gamma_M \operatorname{tr}(\mathbf{H}) \lesssim \epsilon. \tag{33}$$

Now we choose $\mathbb{J} = \{1, 2\}$ so that

$$\begin{aligned}
\frac{D_{\text{eff}}^{\texttt{finetune}}}{M_{\text{eff}}} &\lesssim \frac{1}{M_{\text{eff}}} \cdot \frac{\lambda_1(1 - \gamma_M\lambda_1)^{2N_{\text{eff}}}}{\mu_1} + \frac{1}{M_{\text{eff}}} \cdot \frac{\lambda_2(1 - \gamma_M\lambda_2)^{2N_{\text{eff}}}}{\mu_2} + 0 \\
&\lesssim \epsilon \cdot \frac{\epsilon^2}{\epsilon^2} + \epsilon \cdot \frac{\epsilon^{0.5} \cdot 1}{1} \qquad \text{(by (32))} \\
&\lesssim \epsilon
\end{aligned} \tag{34}$$

For the bias error we have that

$$\begin{aligned}
&\left\| \prod_{t=M}^{M+N-1} (\mathbf{I} - \gamma_t\mathbf{H}) \prod_{t=0}^{M-1} (\mathbf{I} - \gamma_t\mathbf{G})(\mathbf{w}_0 - \mathbf{w}^*) \right\|_{\mathbf{H}}^2 \\
&\leq \lambda_1(1 - \gamma_0\mu_1)^{2M_{\text{eff}}}(1 - \gamma_M\lambda_1)^{2N_{\text{eff}}}(\mathbf{w}^*[1])^2 + \lambda_2(1 - \gamma_0\mu_2)^{2M_{\text{eff}}}(1 - \gamma_M\lambda_2)^{2N_{\text{eff}}}(\mathbf{w}^*[2])^2 \\
&\leq (1 - \gamma_0\mu_1)^{2M_{\text{eff}}}(1 - \gamma_M\lambda_1)^{2N_{\text{eff}}} + \epsilon^{0.5}(1 - \gamma_0\mu_2)^{2M_{\text{eff}}}(1 - \gamma_M\lambda_2)^{2N_{\text{eff}}} \\
&\leq (1 - \gamma_M\lambda_1)^{2N_{\text{eff}}} + \epsilon^{0.5}(1 - \gamma_0\mu_2)^{2M_{\text{eff}}} \\
&\lesssim \epsilon. \qquad \text{(by (32) and (31) )}
\end{aligned} \tag{35}$$

(33), (34) and (35) together imply that $\texttt{ExcessRisk}(\mathbf{w}_{M+N}) \lesssim \epsilon$.

$\square$