# OpenReview forum: "The Power and Limitation of Pretraining-Finetuning for Linear Regression under Covariate Shift"
_NeurIPS.cc/2022/Conference — NeurIPS 2022 Accept_

### Official Review · Reviewer_df3S · 2022-07-06

**Rating:** 5
**Confidence:** 4
**Soundness:** 2 fair
**Presentation:** 3 good
**Contribution:** 2 fair

**Summary:**

This paper investiage the covariate shift problem in a linear regression task. They assume there exits a w* that can simutanously achieves the optimal in both training and testing (target) domain.  They show a model trained on the training domain (with O(N^2) samples) can match the one trained on the target domain (with N samples). Also, they also show the benefits of finetuning on the target domain with limited data. From the technical side, they provide a sharpened upper bound (by a logarithmic factor)  and matched lower bound .

**Questions:**

See Weakness.

**Limitations:**

See Weakness.

**Strengths And Weaknesses:**

Strength:
    1) the paper is well motivated and clearly written.
    2) I didn't check the proof, but the results seem correct to me.
    3) the bounds are sharp and complete.

Major Weakness:
    1) From my perspective, it is strong to assume a w* that is optimal for both domains (or the optimal set of the two domains overlap). This assumption is crucial for the main results such that a model trained with sufficient data in the training domain matches the performance of the one trained on target domain. Given the vast empirical results on the domain adaptation and generalization, there is no evidence that such w* exists or minimizing the training domain loss can lead to testing domain performance gain. I suggest the authors  the provide more supports for the this assumption.  I will change my rate after the rebuttal phase depending on the authors response to this concern.

Minor Weakness:
    1) the notation is dense and the main thereom 3.1 is hard to follow, I suggest the authors to reformulate the results for better readability.
    2) I think (if I am not wrong) the techniques used in this paper is not novel. It is an application of existing methods on the M+N setting.

### Post Rebuttal Comment

After the discussion with the authors, I thinks this paper makes an unreasonable assumption on the training and testing domains share the same optimal $w^*$ in a linear case.  **Though the conditional distribution can be the same in training and testing domain,  the optimal $w$ can still be different due to model misspecification (which is commonly the case in linear models).** In other words, if there is model misspecification, the optimal $w^*$ for the source domain is not optimal for the target domain. By assuming model misspecification, we mean the residual of $y-x^\top w^*$ is not independent of $x$. So let assume $\mathbb{E}_{\mathbb{P}(y|x)}(y-x^\top w^*)=g(x)$, where $g(x)$ is the non-linear part that can't be explained by a linear model.

Denote the source domain with $s$ and target domain with $t$. Because $w^*$ is optimal for the source domain, then we have $\mathbb{E}_s [x (y-x^\top w^*)]=\mathbb{E}_s [x g(x)] = 0$. However, because $P_s(x) \neq P_t(x)$ and further $\mathbb{E}_t [x g(x)] \neq \mathbb{E}_s [x g(x)] $, we have $\mathbb{E}_t [x (y-x^\top w^*) ] = \mathbb{E}_t [x g(x)] \neq 0$.

In conclusion, the $w^*$ optimal for source domain is **NOT** optimal for the target domain due to model misspecification and the shift of $P(X)$.

If one assumes $w^*$ is simutanously optimal for both domains, then we must assume $E_s [x g(x)] = E_t [x g(x)]$ even when $P_s(X) \neq P_t(X)$, which is a prohibitively strong assumption that I have not seen in the existing literature. I think it only holds in rare cases.

Due to this concern that the results of the paper only hold in well specified linear cases, I heavily doubt whether the implication is true for the real applications. I changed score from 6 to 4. I highly suggest the authors to consider different optimal $w^*$ for training and testing domains.

### Post Rebuttal Comment 2

I have read the literature [1]. And it indeed assume the same optimal $w^*$ for both training and testing domains. Though I don't like this assumption and it has problems with model misspecification under distributional shift, my claim "a prohibitively strong assumption that I have not seen in the existing literature" is inaccurate. I raise the score from 4 to 5.

[1] Near-Optimal Linear Regression under Distribution Shift

---

> ### Author Response · Authors · 2022-08-02
> **Response to Reviewer df3S**
>
> **Q1**. “From my perspective, it is strong to assume a w* that is optimal for both domains (or the optimal set of the two domains overlap).”
>
> **A1**. Covariate shift problem, by definition [Sugiyama and Kawanabe, 2012], assumes the source and target distributions differ in their marginal distributions over the input, but coincide in their conditional distribution of the output given the input. For linear regression problems, this turns out to require $\\mathbf{w}\^\*$ to be the same for both source and target domains. So the assumption that $\\mathbf{w}\^\*$ is optimal for both domains simply follows the standard definition of covariate shift.
>
> We agree that for general domain adaptation problems, such $\\mathbf{w}\^\*$ may not exist for both domains, and minimizing the source training loss may not necessarily help minimizing the target test loss. However, for covariate-shift, which is a specific kind of domain adaptation problem where $\\mathbf{w}\^\*$ exists for both domains, we have shown in our paper that, under certain conditions, minimizing the source training loss can lead to small target test loss.
>
> In fact, when such $\\mathbf{w}\^\*$ does not exist on both domains, the problem falls into the category of “model shift”, and we have discussed this more challenging setting as a future work in Section 5 Concluding Remarks.
>
> ---
>
> **Q2**. “the notation is dense and the main thereom 3.1 is hard to follow”
>
> **A2**. Thanks for raising this concern. We guess this is due to the heavy notations defined throughout the paper. We have beautified the displays in Theorem 3.1. We have also added Table 1 in Appendix A to summarize the key notations in Theorem 3.1, Corollaries 3.3 and 3.4, and to make a comparison of the bounds for pretraining-finetuning, pretraining and supervised learning.
>
> ---
>
> **Q3**. “I think (if I am not wrong) the techniques used in this paper is not novel. It is an application of existing methods on the M+N setting.”
>
> **A3**. Our proof technique is indeed developed from some prior works [Zou et al., 2021b, Wu et al., 2021] that are for supervised learning. However, we do not simply adapt these proof techniques from the supervised learning setting to covariate shift setting. In fact, we introduce a new proof technique for analyzing the bias error, which closes the gap between the upper and lower bounds upto constant factors (given bounded signal-to-noise ratio) in [Wu et al., 2021]. As noted in Line 57 and Line 225, our results improve [Wu et al. 2021] even when specialized to the supervised learning setting.

---

> > ### Comment · Reviewer_df3S · 2022-08-03
> > **On the problem of the comment $w*$**
> >
> > Thank you for the reply.
> >
> > Could you point out the exact the part of definition covariate shift in  [Sugiyama and Kawanabe, 2012]? I will theck with it. **Though the conditional distribution can be the same in training and testing domain,  the optimal $w$ can still be different due to model misspecification.** I think the results hold only when assuming no model misspecification in a linear problem, which is too restrictive.
> >
> > Afterall, if a optimal w* exists for both domain, then optimizing in one domain can definitely lead to the gain in the other domain. Then the results are less meaningful for the practice side.
> >
> > As for the technical contribution side, I don't think this contribution is related to (or has something special to do with)  the covariate setting. If I understand correctly, (correct me if I am wrong), the results are an application of the (improved) supervised techniques in the covarite shift setting.

---

> > > ### Author Response · Authors · 2022-08-04
> > > **Thanks for the follow up**
> > >
> > > **Q1**. “Could you point out the exact the part of definition covariate shift in [Sugiyama and Kawanabe, 2012]? I will theck with it. Though the conditional distribution can be the same in training and testing domain. The optimal $w$ can still be different due to model misspecification. I think the results hold only when assuming no model misspecification in a linear problem, which is too restrictive.”
> > >
> > > **A1**. For the definition of covariate shift, please refer to Sections 1.3.3 and 1.3.4 in [Sugiyama and Kawanabe, 2012].
> > >
> > > We indeed assumed a well-specified linear model when drafting our previous response, in which case it is clear that the same conditional probability distribution for both domains implies the same optimal model parameter $\\mathbf{w}\^\*$ for both domains. However, well-specification is not a necessary condition. In particular, one can show that:
> > > $\\mathbf{w}\^\*$ minimizes both $\\mathtt{Risk}\_{\mathtt{source}}(\\mathbf{w})$ and $\\mathtt{Risk}\_{\mathtt{target}}(\\mathbf{w})$ given that
> > > $$
> > > \\mathbb{E}\_{\\mathtt{source}} \\mathbf{x}( y - \\mathbf{x}\^\\top \\mathbf{w}\^\*) = \\mathbb{E}\_{\\mathtt{target}} \\mathbf{x}( y - \\mathbf{x}\^\\top \\mathbf{w}\^\*) = 0.
> > > $$
> > > The above condition can hold for a misspecified linear model. Therefore, assuming the same $\\mathbf{w}\^\*$ for both domains does not exclude model misspecification and is not that restrictive. We thank the reviewer for raising this good point.
> > >
> > > ---
> > >
> > > **Q2**. “Afterall, if a optimal w* exists for both domain, then optimizing in one domain can definitely lead to the gain in the other domain. Then the results are less meaningful for the practice side.”
> > >
> > > **A2**. We agree that in this setting optimizing in one domain can lead to the gain in the other domain. However, how to quantify this performance gain brought by pretraining on the source domain is still a nontrivial question. We have answered this question in Theorem 4.1 by showing that pretraining with $O(N\^2)$ source data can compete with supervised learning using $N$ target data.
> > >
> > > ---
> > >
> > > **Q3**. “As for the technical contribution side, I don't think this contribution is related to (or has something special to do with) the covariate setting. If I understand correctly, (correct me if I am wrong), the results are an application of the (improved) supervised techniques in the covarite shift setting.”
> > >
> > > **A3**. The proof technique is indeed adapted from supervised learning. We improve the proof technique for supervised learning. Based on our improved technique, we further extend it to covariate shift. Such an extension is new and has never been studied in the literature. So we believe our technical contributions include both the improvement and the extension, which are highlighted as follows:
> > >
> > > * Improving existing supervised techniques itself is a solid technical contribution, which requires a very different analyzing strategy (see Appendix D.1).
> > > * The extension from supervised learning to covariate shift is new, which leads to our sharp bounds for pretraining-finetuning, as well as their implications.

---

> > > > ### Comment · Reviewer_df3S · 2022-08-04
> > > > **The problem of model misspecification**
> > > >
> > > > I think if there is model misspecification, the optimal $w^*$ for the source domain is not optimal for the target domain. By assuming model misspecification, we mean the residual of $y-x^\top w^*$ is not independent of $x$. So let assume $\mathbb{E}_{\mathbb{P}(y|x)}(y-x^\top w^*)=g(x)$, where $g(x)$ is the non-linear part that can't be explained by a linear model.
> > > >
> > > > Denote the source domain with $s$ and target domain with $t$. Because $w^*$ is optimal for the source domain, then we have $\mathbb{E}_s [x (y-x^\top w^*)]=\mathbb{E}_s [x g(x)] = 0$. However, because $P_s(x) \neq P_t(x)$ and further $\mathbb{E}_t [x g(x)] \neq \mathbb{E}_s [x g(x)] $, we have $\mathbb{E}_t [x (y-x^\top w^*) ] = \mathbb{E}_t [x g(x)] \neq 0$.
> > > >
> > > > In conclusion, the $w^*$ optimal for source domain is **NOT** optimal for the target domain due to model misspecification and the shift of $P(X)$.
> > > >
> > > > Correct me if I am wrong. Further I suggest the authors to provide more detailed explanations for why the the claim holds: "In particular, one can show that....the above condition can hold for a misspecified linear model."

---

> > > > > ### Author Response · Authors · 2022-08-05
> > > > > **A proof and an example**
> > > > >
> > > > > We think in your reasoning, “$P\_s(x) \\neq P\_t(x)$ implies $\\mathbb{E}\_s[x g(x)] \\neq \\mathbb{E}\_t [x g(x)]$” is not necessarily true. Generally speaking,  we can have two different distributions, but their expectation (or expectation of some function) can be the same.
> > > > >
> > > > > ---
> > > > >
> > > > > We first prove our claim "In particular, one can show that....the above condition can hold for a misspecified linear model." as follows.
> > > > >
> > > > >
> > > > > **Proof of our claim**.
> > > > >
> > > > > Note that $\\mathtt{Risk}\_{\mathtt{source}}(\\mathbf{w})$ is quadratic. Then $\\mathbf{w}\^\*$ minimizes $\\mathtt{Risk}\_{\mathtt{source}}(\\mathbf{w})$ if
> > > > > $$
> > > > > \\nabla \\mathtt{Risk}\_{\mathtt{source}}(\\mathbf{w}) =  \\mathbb{E}\_{\mathtt{source}} \\mathbf{x} (y - \\mathbf{x} \^\\top \\mathbf{w}\^\*) = 0.
> > > > > $$
> > > > > Similarly, $\\mathbf{w}\^\*$ minimizes $\\mathtt{Risk}\_{\mathtt{target}}(\\mathbf{w})$ if
> > > > > $$
> > > > > \\nabla \\mathtt{Risk}\_{\mathtt{target}}(\\mathbf{w}) =  \\mathbb{E}\_{\mathtt{target}} \\mathbf{x} (y - \\mathbf{x}\^\\top \mathbf{w}\^\*) = 0.
> > > > > $$
> > > > > Combining the above two equalities, we see that:
> > > > > $\\mathbf{w}\^\*$ minimizes both $\\mathtt{Risk}\_{\mathtt{source}}(\\mathbf{w})$ and $\\mathtt{Risk}\_{\mathtt{target}}(\\mathbf{w})$ given that
> > > > > $$
> > > > > \\mathbb{E}\_{\\mathtt{source}} \\mathbf{x}( y - \\mathbf{x}\^\\top \\mathbf{w}\^\*) = \\mathbb{E}\_{\\mathtt{target}} \\mathbf{x}( y - \\mathbf{x}\^\\top \\mathbf{w}\^\*) = 0. \\qquad\\qquad\\qquad (*)
> > > > > $$
> > > > >
> > > > > ---
> > > > >
> > > > > In addition, we construct two misspecified linear models on the source and target domains, but the optimal $w^*$ is identical for both domains.
> > > > >
> > > > >
> > > > > **Misspecification example**.
> > > > >
> > > > > Note that condition $(*)$ does not require two random variables $\\mathbf{x}$ and $ y - \\mathbf{x}\^\\top \\mathbf{w}\^\*$ to be independent. For example, consider the following misspecified linear models:
> > > > >
> > > > > * source model
> > > > > $$
> > > > > \\mathbf{x} \\sim \\mathcal{N}(0, \mathbf{G}), y = \\mathbf{x}\^\\top \\mathbf{w}\^\* + \\| \\mathbf{x} \\|_2
> > > > > $$
> > > > >
> > > > > * target model
> > > > > $$
> > > > > \\mathbf{x} \\sim \\mathcal{N}(0, \mathbf{H}), y = \\mathbf{x}\^\\top \\mathbf{w}\^\* + \\| \\mathbf{x} \\|^2_2.
> > > > > $$
> > > > >
> > > > > For the above models, condition $(*)$ holds, because
> > > > > $$
> > > > > \\mathbb{E}\_{\\mathtt{source}} \\mathbf{x}( y - \\mathbf{x}\^\\top \\mathbf{w}\^\*) = \\mathbb{E}\_{\\mathtt{source}} \\mathbf{x} \\| \\mathbf{x} \\|_2 = 0
> > > > > \text{ and }
> > > > > \\mathbb{E}\_{\\mathtt{target}} \\mathbf{x}( y - \\mathbf{x}\^\\top \\mathbf{w}\^\*) = \\mathbb{E}\_{\\mathtt{target}} \\mathbf{x} \\| \\mathbf{x} \\|^2_2 = 0.
> > > > > $$
> > > > > Therefore, $ \\mathbf{w}\^\* $ is the optimal parameter for both source and target models, and both source and target models are misspecified.

---

> > > > > > ### Comment · Reviewer_df3S · 2022-08-05
> > > > > > **The problem of model misspecification**
> > > > > >
> > > > > > I think the author's claim is inaccurate: "Generally speaking, we can have two different distributions, but their expectation (or expectation of some function) can be the same."
> > > > > >
> > > > > > In the covariate shift problem, we consider $P_s(X) \neq P_t(X)$. If you assume $w^*$ is simutanously optimal for both domains, then we must assume $E_s [x g(x)] = E_t [x g(x)]$ even when $P_s(X) \neq P_t(X)$, which is a prohibitively strong assumption that I have not seen in the existing literature. I think it only holds in rare cases.
> > > > > >
> > > > > > Due to this concern that the results of the paper only hold in well specified linear cases, I heavily doubt whether the implication is true for the real applications.
> > > > > >
> > > > > > I will lower my score from 6 to 4.

---

> > > > > > > ### Author Response · Authors · 2022-08-06
> > > > > > > **Response**
> > > > > > >
> > > > > > > We don’t see why our claim "Generally speaking, we can have two different distributions, but their expectation (or expectation of some function) can be the same." is inaccurate. Can you tell us why it is inaccurate? We think this is a basic fact in probability.
> > > > > > >
> > > > > > > In addition, we don’t see why the assumption $E\_s[xg(x)]=E\_t[xg(x)]$ even when $P\_s(X) \\neq P\_t(X)$ is a *prohibitively* strong assumption. Besides the example of misspecified linear models we showed to you in the previous response, we would like to emphasize that this assumption is no stronger than many assumptions made in literature. Please see the following examples.
> > > > > > >
> > > > > > > ---
> > > > > > > **[Bartlett et al. 2020]**
> > > > > > >
> > > > > > > Since you mentioned the benign-type results, let us take a closer look at the seminal work by [Bartlett et al. 2020], and show that their linear model assumption implies a shared $\\mathbf{w}\^\*$ in the covariate shift setting.
> > > > > > >
> > > > > > > In particular, in the Definition 1 (Bullet 4) of Bartlett et al 2020, they assumed a zero mean additive noise in their linear model, i.e.,
> > > > > > > $$
> > > > > > > \\mathbb{E} [ y | \\mathbf{x} ] = \\mathbf{x}\^\\top \\mathbf{w}\^\*, \\text{ for } \\mathbf{w}\^\* \\in \\text{argmin } \\mathbb{E} [(y-\\mathbf{x}\^\\top \\mathbf{w})\^2 ].
> > > > > > > $$
> > > > > > > Let us consider such linear models in the covariate shift setting, i.e.,
> > > > > > > * zero mean noise condition for both domains:
> > > > > > > $
> > > > > > > \\begin{aligned}
> > > > > > > \\mathbb{E}\_{source}[ y | \\mathbf{x}] = \\mathbf{x}\^\\top \\mathbf{w}\^\*\_{source}, \\text{ for } \\mathbf{w}\^\*\_{source} \\in \\text{argmin } \\mathbb{E}\_{source} [(y-\\mathbf{x}\^\\top \\mathbf{w})^2], \\\\
> > > > > > > \\mathbb{E}\_{target}[ y | \\mathbf{x}] = \\mathbf{x}\^\\top \\mathbf{w}\^\*\_{target}, \\text{ for } \\mathbf{w}\^\*\_{target} \\in \\text{argmin } \\mathbb{E}\_{target} [(y-\\mathbf{x}\^\\top \\mathbf{w})^2],
> > > > > > > \\end{aligned} \\qquad (1)
> > > > > > > $
> > > > > > > * covaraite shift condition:
> > > > > > > $
> > > > > > > \\mathbb{P}\_{source} (y| \\mathbf{x}) = \\mathbb{P}\_{target}(y|\\mathbf{x}). \\qquad (2)
> > > > > > > $
> > > > > > >
> > > > > > > By (2) we see that
> > > > > > > $
> > > > > > > \\mathbb{E}\_{source}[ y | \\mathbf{x}] = \\mathbb{E}\_{target}[ y | \\mathbf{x}],
> > > > > > > $
> > > > > > > and then by $(1)$ we have $\\mathbf{x}^\top \\mathbf{w}\^\*\_{source} = \\mathbf{x}\^\\top \\mathbf{w}\^\*\_{target}$ for every $\\mathbf{x}$. This implies $\\mathbf{w}\^\*\_{source} = \\mathbf{w}\^\*\_{target}$.
> > > > > > >
> > > > > > > Therefore our assumption, a shared $\\mathbf{w}^*$ for both source and target domains, is a natural generalization of the model assumption in [Bartlett et al. 2020] under covariate shift. The work by [Bartlett et al. 2020] has improved the understanding of benign overfitting in the community. We think our result will also improve the understanding of pretraining-finetuning and have a great implication for real applications.
> > > > > > >
> > > > > > > ---
> > > > > > > **Covariate shift literatures**
> > > > > > >
> > > > > > > In addition, there are several other papers on covariate shift that make this assumption. For example, [Lei et al. 2021] also studied the covariate shift problem under the assumption that $\\mathbf{w}\^\*$ is shared for both source and target domains. [Ma et al. 2022] assumed the above condition $(1)$, so the $\\mathbf{w}\^\*$ is shared between the source and target domain in the covariate shift setting. Our assumption is no stronger (or weaker) than theirs.
> > > > > > >
> > > > > > > ---
> > > > > > >
> > > > > > >
> > > > > > > [Bartlett et al. 2020] Bartlett, Peter L., et al. "Benign overfitting in linear regression." Proceedings of the National Academy of Sciences 117.48 (2020): 30063-30070.
> > > > > > >
> > > > > > > [Lei et al. 2021] Lei, Qi, Wei Hu, and Jason Lee. "Near-optimal linear regression under distribution shift." International Conference on Machine Learning. PMLR, 2021.
> > > > > > >
> > > > > > > [Ma el al. 2022] Ma, Cong, Reese Pathak, and Martin J. Wainwright. "Optimally tackling covariate shift in RKHS-based nonparametric regression." arXiv preprint arXiv:2205.02986 (2022).

---

> > > > > > > > ### Comment · Reviewer_df3S · 2022-08-06
> > > > > > > > **Clarification**
> > > > > > > >
> > > > > > > > Your claim "Generally speaking, we can have two different distributions, but their expectation (or expectation of some function) can be the same." can be true in some cases. But more general, they don't hold in practice or you need to make ASSUMPTION on the same of expectations between two domains. Indeed, you give an example it hold in a case, however, I can easily give an example that this not holds, i.e., $g_s(x)=x^3$. How can you guarantee $E_s[xg(x)]=E_t[xg(x)]$? If this only holds for SOME cases, you need to make assumptions or point out under with cases this hold. Then it will be a seperate issue to discuss.
> > > > > > > >
> > > > > > > > The results of [Bartlett et al. 2020] is for the iid setting. It can also easily extended to model misspecified setting in the iid case. In the distributional shift case, it is fundamentally different.
> > > > > > > >
> > > > > > > > I have spent too much time in this discussion and will no longer response. I think AC will make a fair decision.

---

> > > > > > > > > ### Author Response · Authors · 2022-08-06
> > > > > > > > > **Thank you for spending the time discussing this with us**
> > > > > > > > >
> > > > > > > > > In our humble opinion, even with the well-specified linear model assumption, our results are still significant and shed great light on understanding pretraining-finetuning under the covariate shift. While pretraining-finetuning has been widely used in deep learning, without a solid understanding in the simplest possible linear regression setting, there is little hope to have a deep understanding of pretraining-finetuning in the more general setting.

---

> > > > ### Comment · Reviewer_df3S · 2022-08-04
> > > > **Another question**
> > > >
> > > > This paper use the benign-style techniques and results. I am curious on why not using the classical OLS techniques, e.g., those in [1]. Is there something special to use the benign  techniques for the covariate shift problem? Or what may be the difference between the results if we instead use classical OLS techniques [1] to analyze this problem.
> > > >
> > > > [1] Daniel HSU, Sham Kakade, Tong Zhang, Random Design Analysis of Ridge Regression.

---

> > > > > ### Author Response · Authors · 2022-08-05
> > > > > **Thanks for the question**
> > > > >
> > > > > For pretraining-finetuning with OLS or ridge regression, it is possible to use techniques from [1] to show a risk upper bound. However, it is not clear how to derive an instance-wise matching lower bound based on techniques from [1]. With only an upper bound, we cannot do an instance-wise comparison between the sample efficiency of pretraining and that of supervised learning like our Theorem 4.1.
> > > > >
> > > > > In addition, we would like to emphasize that even though our proof techniques originate from works studying benign overfitting of SGD [Zou et al., 2021b, Wu et al., 2021], our results also cover the classical regime as a special case.
> > > > >
> > > > > [1] Daniel HSU, Sham Kakade, Tong Zhang, Random Design Analysis of Ridge Regression.

---

### Official Review · Reviewer_XiiA · 2022-07-10

**Rating:** 4
**Confidence:** 4
**Soundness:** 3 good
**Presentation:** 3 good
**Contribution:** 3 good

**Summary:**

This paper investigates a transfer learning approach with pretraining on the source data and finetuning based on the target data and establishes sharp instance-dependent excess risk upper and lower bounds for this approach.

**Questions:**

The regression problem is very old. What is the factual applications of the theoretical analysis?

**Limitations:**

The limitations of the presented analysis could be discussed in more detail.

**Strengths And Weaknesses:**

### Originality

The paper's idea is very novel. To the best of my knowledge, the theoretical analysis of the transfer learning paradigm remains unclear.

### Quality

The technical part of the paper seems correct as far as I can see.

### Clarity

The paper is well-written.

### Significance

The question studied in the paper is significant. Understanding the power and limitation of pretraining-finetuning for linear regression under covariate shift is meaningful.

---

> ### Author Response · Authors · 2022-08-02
> **Response to Reviewer XiiA**
>
> Thank you for recognizing the originality, quality, clarity and significance of our work.
>
> ---
>
> **Q1**. “The regression problem is very old. What is the factual applications of the theoretical analysis?”
>
> **A1**. We respectfully disagree with your comment. Linear regression is an old but classical machine learning model. Yet even for linear regression, there is no such kind of result before our work on pretraining and finetuning. Without a full understanding about the simplest possible linear regression, it seems unlikely that one can theoretically understand pretraining and finetuning for more general machine learning models. As our work is the first work on studying problem-dependent bounds for pretraining and finetuning, we believe we have taken an important step towards this direction and the contribution of our work is significant.

---

### Official Review · Reviewer_5qgQ · 2022-07-11

**Rating:** 7
**Confidence:** 3
**Soundness:** 4 excellent
**Presentation:** 4 excellent
**Contribution:** 4 excellent

**Summary:**

This paper presents a theoretical study of the generalization performance of the pretraining-finetuning method with linear regression setting under covariate shift. Especially, this paper derives the upper and lower bounds for the generalization achieved by the pretraining-finetuning method. With the bounds, this paper characterizes the power of the pretraining, the pretraining-finetuning method, and the limitation of the pretraining compared to supervised learning.

**Questions:**

In line 263, "middle" eigenvalues subspace of H falls into the top eigenvalues subspace of G. Can you explain in more detail?

**Ethics Review Area:**

["I don’t know"]

**Limitations:**

The paper seems fine to me in this regard.

**Strengths And Weaknesses:**

This paper is interesting and well-written. I think that this is a good paper that should be accepted. One minor concern is how standard Assumption1 is. If the assumption is standard or applies to any common distributions under covariate shift, it would be better to state it with some examples.

It would be better to summarize the bounds by showing a detailed comparison between only pretraining, pretraining-finetuning, and finetuning.

---

> ### Author Response · Authors · 2022-08-02
> **Response to Reviewer 5qgQ**
>
> Thanks for recognizing the contribution of our paper.
>
> ---
>
> **Q1**. “One minor concern is how standard Assumption1 is. If the assumption is standard or applies to any common distributions under covariate shift, it would be better to state it with some examples.”
>
> **A1**. In our submission, we have already given two examples in the discussion after Assumption 1. Here we give more examples, which are originally from  [Zou et al., 2021b, Wu et al., 2021]:
>
> Assumption 1A  is satisfied for data distributions with a bounded kurtosis along every direction [3], i.e., there is a constant $\\alpha>0$ such that
> $$
> \\text{for every}\\ \\mathbf{v}, \\mathbb{E}\_{source} [ \\langle \\mathbf{v}, \\mathbf{x} \\rangle\^4] \\le \\alpha \\cdot \\langle \\mathbf{v}, \\mathbf{G} \mathbf{v} \\rangle\^2, \\text{ and }
> \\mathbb{E}\_{target} [ \\langle \\mathbf{v}, \\mathbf{x} \\rangle\^4] \\le \\alpha \\cdot \\langle \\mathbf{v}, \\mathbf{H} \mathbf{v} \\rangle\^2.
> $$
>
> ---
>
> **Q2**. “It would be better to summarize the bounds by showing a detailed comparison between only pretraining, pretraining-finetuning, and finetuning.”
>
> **A2**. Thank you for your suggestion. We have added Table 1 (in Appendix A) that compares the bounds of pretraining, pretraining-finetuning, and finetuning in the revision. In the camera ready version, when we are allowed to have one more page, we can move this table to the main text.
>
> ---
>
> **Q3**. “In line 263, "middle" eigenvalues subspace of H falls into the top eigenvalues subspace of G. Can you explain in more detail?”
>
> **A3**. Note that Theorem 4.2 only requires $\\| \\mathbf{H}\_{\\mathbb{K}\^\\dagger} \\|\_{\\mathbf{G}}$ to be upper bounded, i.e., the eigenvalues of $\\mathbf{H}$ belong to the set $\\mathbb{K}\^\\dagger$ mostly fall into the top eigenvalues subspace of $\\mathbf{G}$. The index set $\\mathbb{K}\^\\dagger$ captures the so-called “middle” eigenvalues subspace.
>
> ---
>
> [3] Dieuleveut, Aymeric, Nicolas Flammarion, and Francis Bach. "Harder, better, faster, stronger convergence rates for least-squares regression." The Journal of Machine Learning Research 18.1 (2017): 3520-3570.

---

### Official Review · Reviewer_a3cN · 2022-07-12

**Rating:** 7
**Confidence:** 3
**Soundness:** 4 excellent
**Presentation:** 4 excellent
**Contribution:** 3 good

**Summary:**

This paper studies the performance of one-pass SGD when pre-trained on a large linear regression source task and fine-tuned on a smaller linear regression target task with a mild degree of covariate shift. In this setting, O(N^2) data points from the source data are as effective as O(N) data points from the target task. The study is mostly theoretical based on a bias-variance decomposition where the bias term is due to limited iterations of the single-pass algorithm and the variance term is due to noise. An experiment verifies the results on a synthetic problem.

**Questions:**

See weaknesses section

**Limitations:**

See weaknesses section

**Strengths And Weaknesses:**

The paper is well written and well organized. The proofs are in the appendix which was not checked but the arguments in the main paper are intuitive and convincing. The choice of algorithm and the domain shift make the findings less surprising as I will elaborate below, but the theoretical characterization in the paper may pave to way for more interesting results. Altogether I am voting for acceptance but I strongly recommend highlighting the following limitations in the introduction.

1. All the results are on a quite benign form of covariate shift where the model reaches the same solution on both the source and the target task. Even if no data point is available from the target task, the model can reach zero error by training on the first task. This is where the power of pretraining largely comes from. The question is then how much training on the source task can slow down the optimization or hurt generalization compared to training on the target task.

2. The studied algorithm is one-pass SGD. Optimization and generalization of this algorithm are connected together in a way that the insights from this paper may not extend to the more commonly used multi-pass SGD. The number of data points in one-pass SGD determines the number of iterations and how far the parameters can move towards the optimal solution. The limitation of pretraining appears to largely come from the fact that reducing the loss on the target task requires more iterations if training is on the source task instead of the target task. In single-pass SGD, since the number of iterations is equal to the number of data points, more iterations means more data points. Multi-pass SGD with limited data points can make up for this slow convergence with more epochs even if the number of data points is limited. I am not sure then if these findings on one-pass SGD about the number of required data points extend to multi-pass SGD. I suggest clarifying in the intro and abstract that this is a study of the performance of one-pass SGD under covariate shift and using applications of one-pass SGD in ML to highlight the significance.

Minor comments:

(i) Line 241 states that a bounded signal-to-noise ratio is of statistical interest. Is the ratio bounded from above here? And could you provide citations for the claim about statistical interest?

Typos:
- Line 262: regarding
- Line 273: Supervised
-----------------------------
Update: The revision addressed my second comment and so I increased the score to 7. As for the first comment, although this assumption limits the applicability of the current result, I think analyzing the behavior of single-pass SGD under this assumption is still valuable.

Note on clarity: Covariate shift has a rather standard definition in machine learning: The two tasks have the same conditional distribution p(y|x) but different marginal distribution p(x). See "key assumption 1" in [1] for example. As the reviewer df3S explained, if there is a chance of model misspecification, the assumption that the two tasks share the same optimal parameters does not follow from this definition. (An elaborate discussion is provided in [2].) Currently, the submission is including the assumption of shared optimal parameters in the definition of covariate shift. Even if some previous papers have defined covariate shift in this way, doing this would confuse a majority of readers who expect the standard definition of covariate shift. I strongly recommend separating this assumption from the definition of covariate shift and editing the corresponding statements in the paper to help with readability. One example is the sentence "For covariate shift problem, pretraining with infinite many source data can learn the true model" in line 230 in the revision.

[1] Huang, Jiayuan, et al. "Correcting sample selection bias by unlabeled data." NeurIPS, 2006

[2] Shimodaira, Hidetoshi. "Improving predictive inference under covariate shift by weighting the log-likelihood function." Journal of statistical planning and inference, 2000

---

> ### Author Response · Authors · 2022-08-02
> **Response to Reviewer a3cN**
>
> Thanks for recognizing our contribution. We have fixed the typos in our revision.
>
> ---
>
> **Q1**. “All the results are on a quite benign form of covariate shift where the model reaches the same solution on both the source and the target task. Even if no data point is available from the target task, the model can reach zero error by training on the first task. This is where the power of pretraining largely comes from. The question is then how much training on the source task can slow down the optimization or hurt generalization compared to training on the target task.”
>
> **A1**. We totally agree. Under covariate shift, the model can reach zero excess risk by training with zero target data and infinite source data. However, when there are only a finite number of source data and target data, how much training on the source task is comparable to training on the target task is still a nontrivial question. We have answered this question in Theorem 4.1 in our main results. We have emphasized this in the revised paper (see Line 230).
>
> ---
>
> **Q2**. “The studied algorithm is one-pass SGD. Optimization and generalization of this algorithm are connected together in a way that the insights from this paper may not extend to the more commonly used multi-pass SGD. … The limitation of pretraining appears to largely come from the fact that reducing the loss on the target task requires more iterations if training is on the source task instead of the target task. In single-pass SGD, since the number of iterations is equal to the number of data points, more iterations means more data points. Multi-pass SGD with limited data points can make up for this slow convergence with more epochs even if the number of data points is limited. I am not sure then if these findings on one-pass SGD about the number of required data points extend to multi-pass SGD. ...”
>
> **A2**. Thank you for your suggestion. Indeed the results in our paper are for online SGD, and may not be directly extendable to multi-pass SGD, due to the different optimization and generalization behaviors of online SGD and multi-pass SGD. We have emphasized this in the abstract and introduction in the revision.
>
> ---
> **Q3**. “Line 241 states that a bounded signal-to-noise ratio is of statistical interest. Is the ratio bounded from above here? And could you provide citations for the claim about statistical interest?”
>
> **A3**. Yes. The signal-to-noise ratio (SNR) is related to the “Coefficient of determination” ($R\^2$) (see, e.g., Section 11.3.4 in [1])，defined as
>
> $$
> R^2 = \\frac{1}{(1+1/\\mathtt{SNR})} = 1 - \\frac{\\mathtt{risk}(\\mathbf{w}\^*)}{\\mathtt{risk}(0)}
> $$
>
> So $\\mathtt{SNR} < \infty$ is equivalent to $R^2 < 1$, which means that $\sigma^2 > 0$ and the data cannot be perfectly fitted with a linear model [1]. In this regime, there are non-trivial excess risk lower bounds for Gaussian linear regression (see, e.g., [2]). On the contrary, if $\\mathtt{SNR}  = \infty$ or $R^2 = 1$, then $\sigma=0$; in this regime one solves the problem by fitting all data, which is essentially an optimization problem.
>
> ---
>
> [1] Casella, George, and Roger L. Berger. Statistical inference. Cengage Learning, 2021.
>
> [2] Breiman, Leo, and David Freedman. "How many variables should be entered in a regression equation?." Journal of the American Statistical Association 78.381 (1983): 131-136.

---

### Meta-Review · Area_Chair_esgK · 2022-08-26

**Recommendation:** Accept
**Confidence:** Less certain

**Metareview:**

After the rebuttal, the majority of the reviewers were convinced that the paper is novel and interesting and it should be accepted. In preparing the camera-ready, I suggest to the authors to take into account the reviewers' comments to better explain the used assumptions to the readers and avoid potential misunderstandings.

**Award:**

No

---

### Decision · Program_Chairs · 2022-09-14

Accept